# Passive margins in accreting Archaean archipelagos signal continental stability promoting early atmospheric oxygen rise

Yaying Peng[1], Timothy Kusky [1,2] ✉, Lu Wang [1] ✉, Zhikang Luan [1], Chuanhai Wang[1], Xuanyu Liu [1], Yating Zhong[1] & Noreen J. Evans [3]

Significant changes in tectonic style and climate occurred from the late Archaean to early Proterozoic when continental growth and emergence provided opportunities for photosynthetic life to proliferate by the initiation of the Great Oxidation Event (GOE). In this study, we report a Neoarchaean passive-margin-type sequence (2560–2500 million years ago) from the Precambrian basement of China that formed in an accretionary orogen. Tectonostratigraphic and detrital zircon analysis reveal that thermal subsidence on the backside of a recently amalgamated oceanic archipelago created a quiet, shallow water environment, marked by deposition of carbonates, shales, and shallow water sediments, likely hosts to early photosynthetic microbes. Distinct from the traditional understanding of passive margins generated by continental rifting, post-collisional subsidence of archipelago margins represents a novel stable niche, signalling initial continental maturity and foreshadowing great changes at the Archaean-Proterozoic boundary.

Earth experienced numerous major developments around the Archaean-Proterozoic transition, including the appearance of continents[1] and the remarkable change in surface environments marked by the Great Oxidation Event (GOE), but their relationship to underlying tectonic mechanisms has been a contentious issue for decades. The early Archaean may have been characterized by numerous oceanic arcs, largely submerged, and many independent data sets suggest that these arcs grew and amalgamated into emergent continents sometime in the late Archaean[1,2]. It has been proposed that the stabilization of continents greatly affected surface biological activities that facilitated oxygen accumulation in the atmosphere[3]. Sedimentary basins form in response to horizontal and/or vertical tectonic movements, recording critical information about tectonic style in ancient orogens and about Earth surface conditions[3–6], thus can yield data about these spectacular changes in our planet's environment and the underlying causal mechanisms.

The North China Craton (NCC) is divided into an old Eastern Block with magmatic ages of 2.9–2.55 Ga and isolated fragments up to 3.8 Ga[7], the Neoarchaean Central Orogenic Belt (COB), the Archaean Western Block, and the Inner Mongolia-Northern Hebei Orogen that progressively accreted to the Eastern Block in the Paleoproterozoic (Fig. 1a)[8–10]. These accretionary events culminated in NCC's collision with the Columbia Supercontinent at 1.85 Ga, causing widespread high grade metamorphism and Tibetan-style crustal thickening[8–11].

The Eastern Block (EB) of the NCC was previously considered to consist of a series of microblocks of older tonalite-trondhjemite-granodiorite (TTG) crust with intervening greenstone belts that converged causing deformation and metamorphism between 2.7–2.55 Ga to form protocontinental nucleii[7,12–14]. These were later suggested to be a southwest Pacific-like archipelago of arcs and microcontinents that accreted between 2.7–2.55 Ga, forming a protocontinental block in the east, upon which successive arcs and other terranes were accreted from 2.52 to 1.85 Ga[8,15]. The COB contains the boundary between older amalgamated arc terranes to the east, and ca. 2.7–2.52 Ga intra-oceanic arcs that collided with the EB by ca. 2.5 Ga (Fig. 1a)[16,17], closing a preexisting ocean, vestiges of which are marked by ophiolitic mélange,

[1]State Key Lab of Geological Processes and Mineral Resources, Center for Global Tectonics, School of Earth Sciences, China University of Geosciences, Wuhan, China. [2]Badong National Observatory and Research Station for Geohazards, China University of Geosciences, Wuhan, China. [3]School of Earth and Planetary Sciences, John de Laeter Centre, Curtin University, Perth, WA 6845, Australia. ✉e-mail: tkusky@gmail.com; wanglu@cug.edu.cn

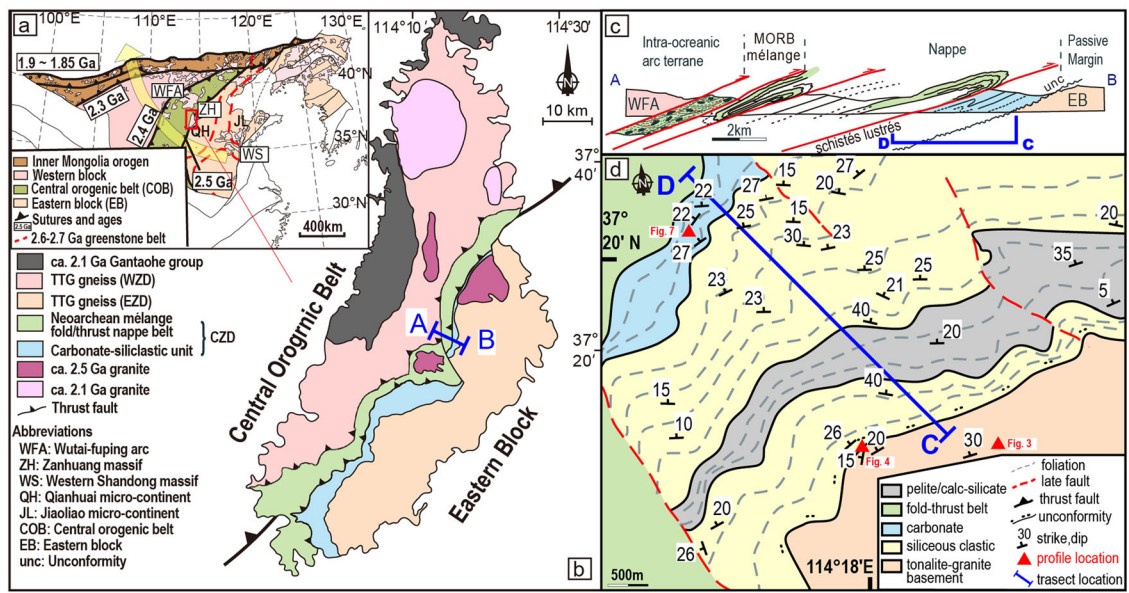

**Fig. 1 | Geological maps of study area. a** Map of North China Craton (after[8]) showing progressive growth by collisional tectonics. See text for explanation. The Zanhuang Complex is shown in red box. **b** Geological map of the Zanhuang Complex (after[16,25,30]), location marked in **a**. See text for geological setting. Blue line marks the location of the profile section (Section A–C) with details shown in **c**. **c** Profile section showing the ophiolitic mélange[16], Alpine-style fold-thrust-belt[17,21], and passive margin sequence of this study (transect C-D, in blue line). **d** Geological map of study area (adapted from[24]). Red triangles show location of mapped profiles.

fore-arc magmatic rocks, far-travelled sub-horizontal nappes, eclogitic oceanic crust, and paired metamorphic belts[17–19], formed at the Archaean-Proterozoic boundary.

The Zanhuang massif straddles the COB/EB boundary (Fig. 1a), and is divided into the Western, Eastern, and Central Zanhuang Domains (WZD, CZD, and EZD) (Fig. 1b). The WZD is part of the Wutai/Fuping intraoceanic arc terrane, and the EZD is located on the western edge of the EB protocontinent[8]. In between, the CZD includes an ophiolitic tectonic mélange and Alpine-style nappes thrust over a marble-siliciclastic unit attached to the EZD at ca. 2500–2520 Ma[16,17,20–22]. Thus, the CZD delineates a Neoarchaean suture where the intra-oceanic arc terrane in the COB was thrust over a marble-siliciclastic unit resting unconformably over the western margin of the EB protocontinent (Fig. 1c, d). These rocks were originally mapped as the Neoarchaean Fangjiapu Formation in the Zanhuang Group (or the Tuanpokou Formation in the Fuping Group)[23,24], but in following sections, we use a lithostructural nomenclature based on our new results. Parts of the Western Zanhuang Domain are unconformably overlain by flat-lying sedimentary and volcanic rocks of the 2.1 Ga Gantaohe Group[22,23] (Fig. 1b) showing that major deformation in the Zanhuang massif was over by that time.

Here, we report a well-preserved Neoarchaean passive-margin-type shallow water sequence from the Precambrian basement of northern China, in the North China Craton (NCC), documenting its sedimentary facies, structures, and detrital zircon geochronology. We use this data to elucidate the hitherto incomplete knowledge of the tectonic history of NCC's protocontinental Eastern Block. Based on our study, we suggest that the earliest passive-margin-like sedimentary basins are a key indicator of initial stability of actively forming cratons at the Archaean-Proterozoic boundary and the onset of the GOE.

## Results

### Tectonostratigraphy

Detailed structural and stratigraphic relationships between the EB, the overlying autochthonous (meaning resting in the place it was deposited) sedimentary sequence, and a series of allochthonous (meaning displaced from the location which it formed) Alpine-style fold nappes are well-exposed in the Zanhuang massif. We describe a stratotype section (transect C-D, Figs. 1c, d, 2) divided, from base to top, into 4 units: (1) a tonalitic-granitic autochthonous basement gneiss complex, (2) the autochthonous coastal siliceous sediments, (3) the autochthonous shallow marine platformal sediments, and (4) the allochthonous units.

The basement (Unit 1 in Fig. 2; Fig. 3) is variably well-exposed in the EZD, with only basic geological maps previously available, suggesting that the EZD is generally composed of tonalite-trondhjemite-granite[23–25]. Our targeted detailed litho-structural mapping in key locations better constrains structural and temporal relationships between units in the basement and cover (Fig. 3). Outcrops near the base of the stratosection are dominated by products of partial melting of older crust and magmatic activities, including monzogranite and later intrusions of quartz diorite (Fig. 3a, b). The crosscutting relationships in less melted regions show that monzogranite cuts tonalite, and they both are deformed into isoclinal folds (Fig. 4a, c); hence, the monzogranite is interpreted as syn-deformational, or intruded close to the late stage of deformation. The quartz diorite is preserved as individual diffuse bodies sparsely distributed in the EZD, characteristic of mid-crust intrusions. These intrusions (Fig. 3) crosscut the more widely distributed monzogranite, which is slightly younger than the monzogranite (Supplementary Fig. 1, detailed data can be found in Supplementary Dataset 1). Since the quartz diorite is also deformed with the monzogranite and the basement, they are interpreted to be similarly intruded in the late stage of deformation. Data presented below, shows that the basement complex is clearly older than the cover, but was remobilized at the time of collision.

A sequence of silicious (meta)sedimentary rocks (Unit 2 in Fig. 2) rests unconformably on the tonalitic-granitic basement (Unit 1 in Fig. 2; Fig. 5a). The contact is deformed (see next section; Fig. 4), yet the unconformable relationship is retained. Overlying (meta)sediments preserve deformed, yet well-developed bedding defined by variations in composition and grain size. Unit 2 includes four subunits grading up from (i) transgressive (meta)sandstone, (ii) layered (meta)sandstone (Figs. 2a, 5a), (iii) pebbly (meta)conglomerate (Fig. 5b), and (iv) (meta)greywacke (Fig. 5c). The transgressive sandstone subunit directly

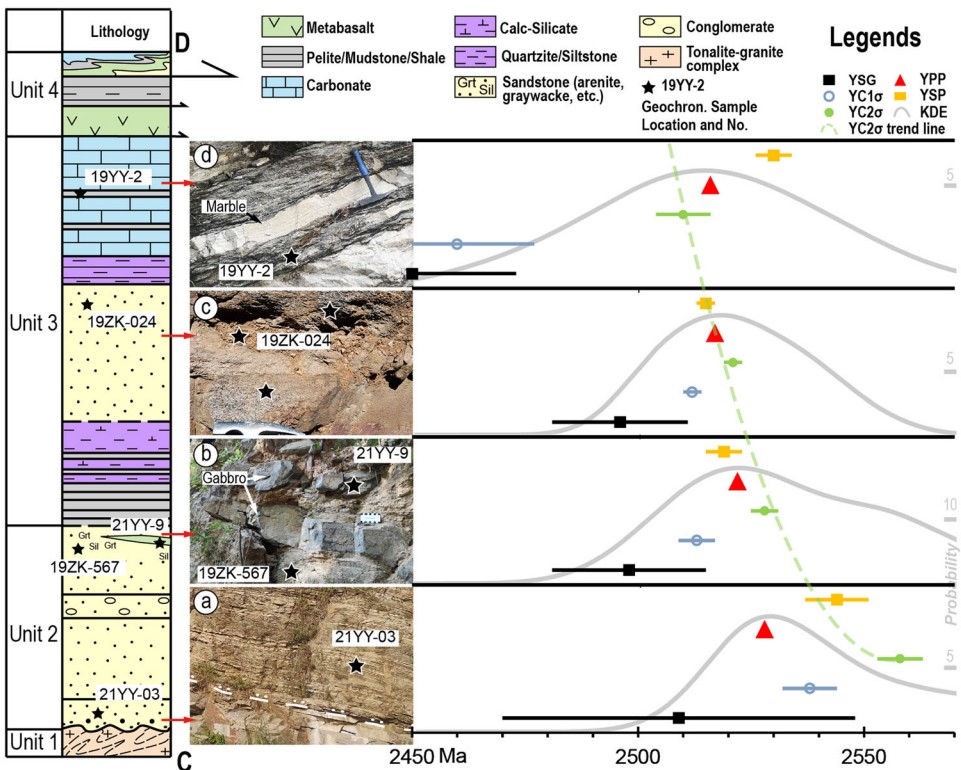

**Fig. 2 | Tectono-stratigraphic column and maximum depositional age. a–d** Field pictures of rock units sampled for geochronology of transect C-D. Sample locations are shown on the stratigraphic column, and on the accompanying photos. Right side shows plots of age calculations from detrital zircon data, and trend line (not quantified). In (**a**) dashed line with dots shows the unconformity surface, black stars in all photos show zircon sample locations. **b** Amphibolitic meta-gabbroic sill (sample 21yy-9, see text) cutting metasandstone of Unit 2. **c** Metasandstone of Unit 3. **d** Interbedded carbonate (marble) and metapelite. YSG youngest single grain. YC1σ/2σ youngest single cluster overlapping at 1σ/2σ uncertainty. YPP youngest probability peak in mode of kernel density estimation[65]. YSP youngest statistical population[39]. See text for methods description.

covers the unconformity, preserving a poorly sorted basal pebbly sandstone (Fig. 6a), and grading up into layered sandstone (Fig. 5b). A pebbly conglomerate unit lies above, with cm sized pebbles akin to the tonalitic basement (Fig. 5b). The conglomerate is overlain by grey-wacke (Fig. 2b) composed of quartz and biotite, locally with garnet porphyroblasts (Fig. 5c) and sillimanite, concentrated in subunits with a high matrix content. Late gabbroic intrusions cutting the greywacke are deformed with the regional strain field (Fig. 2b). Generally, the siliciclastic rocks from Unit 2 are from a high-energy transgressive setting, consistent with a near-shore environment.

Unit 3 is divided into lower finely laminated metapelite-calcsilicate-quartzite (Figs. 2, 5d, e) and higher thickly layered meta-arenite (Fig. 2c), quartzite (Fig. 5f), and marble-metapelite (Fig. 5g). The former consists of interlayered mica schist, quartzite, and calc-silicates, with shallow water mudstone, quartz siltstone and micrite protoliths. The unit shows a transition from more metapelite at the base (Fig. 5d) to dominant quartzite upwards (Fig. 5e), while calcsilicates are interlayered throughout the unit. The higher part of Unit 3 comprises sediments from low energy environments (Fig. 2). Thick units of arenite (Fig. 2c) form strongly foliated layers of quartz and feldspar with minor biotite. The arenite is covered by thin layers of whitish, greyish, brownish, and greenish quartzite (Fig. 5f), with minor compositional differences. Upwards, it grades into strongly deformed marble-metapelite, with inter-layered marble and grey to black mica schist (Fig. 2d). Some marble layers reach several metres in thickness even after deformation (Fig. 5g). The gradual transition from mudstone (metapelite) to siltstone (quartzite) and limestone (marble) indicates increasing depositional stability and sediment maturity during deposition. Notably, the occurrence of micrite (calcsilicate) and limestone

(marble) strongly suggests a transition into a quiet shelf environment allowing precipitation of carbonates.

Unit 4 is formed by allochthonous slices of metabasalt (Figs. 2, 5h), metasiltstone (Fig. 5i), and nappes thrust upon the metasediments, where a mica rich shear zone acts as the decollement, whose detrital zircons constrain a maximum deformation and emplacement age of 2520 Ma[17]. Contacts between all lithological units are highly sheared, with intense deformation, yet the lithological sequence beneath the decollement is largely intact.

In summary, the autochthonous sequence records a transgressive setting from coastal (Unit 2) to shallow marine platform facies (Unit 3), deposited over a granitic-tonalitic gneiss of a microcontinental base-ment (Unit 1). The allochthon (Unit 4), overthrust the autochthonous sequence, and closed a pre-existing ocean by 2500 Ma[26].

## Structural relationships

The Zanhuang autochthon and associated units include three main structural units: basement (Unit 1), autochthonous metasedimentary units (Unit 2, 3), and overthrust allochthonous units (Unit 4 and higher segments). Here we present structural studies covering the poly-genetic basement, the unconformity between the basement and the metasediments, the autochthonous metasediments, with specific focus on the well-exposed stratotype section, and compare them to previous research on the allochthon, mélange and nappes, illustrating the chronological relationships between units, and their meaning in terms of regional tectonic evolution.

Our structural analysis (Fig. 3) suggests that the basement underwent more deformation events than the overlying sequence. Structural analysis shows that both the foliation and lineation have an extra fabric element, in addition to the main NNW-dipping foliation

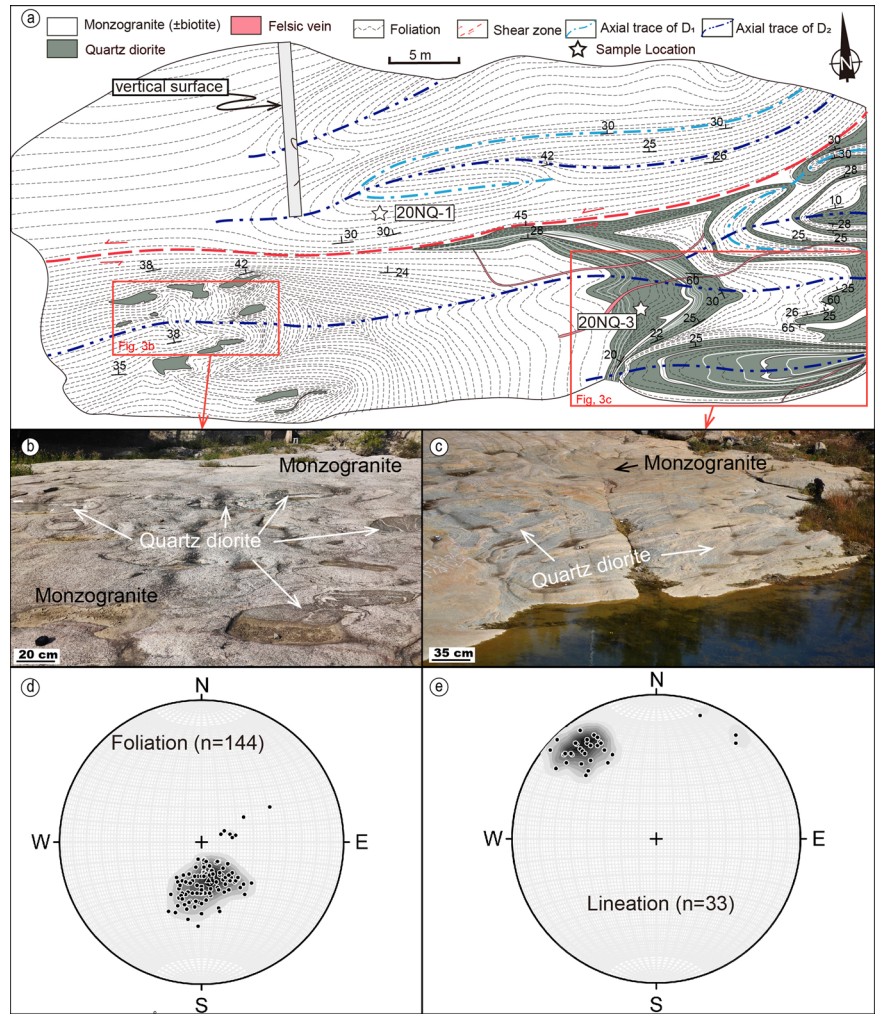

**Fig. 3 | Detailed structural map of the Xiashiliu River exposure of the basement complex of the profile.** The location of this out crop is shown in Fig. 1d. **a** Structural map of an exposure of the multi-component basement complex, where three episodes of deformation are recognized with different traces of fold axes. Mapping was performed at 1:50 scale, using a grid-line technique and vertical drone-hosted photo-imagery. **b, c** Field photos of the exposure, with white dashed lines outlining the location of quartz diorite. **d** Poles of foliation of the exposure, showing some dataset apart from the main concentration. **e** Mineral lineation, showing an extra concentration point from the main concentration.

and NW plunging lineation, that has elsewhere been related to the circa 2.5 Ga arc-continent collision in the Central Orogenic belt[16,17,21]. Widespread anatexis and intense reworking of the basement makes the formation ages difficult to extract. There is a paucity of previously reported ages of the Archaean rocks in the basement igneous complex, limited to a few metamorphic age reports of 2.51–2.46 Ga[27,28], constraining that the basement formation predates this metamorphism. Our study constrains the anatectic event from monzogranite to be 2489 ± 6 Ma (Supplementary Fig. 1b) and from the quartz diorite to 2486 ± 4 Ma (Supplementary Fig. 1c), while the inherited basement xenocrysts indicate older basement ages of 2580-2800 Ma (Supplementary Fig. 1a, b).

Our mapping shows that the siliclastic sediments are in unconformable contact with the underlying tonalite-granite complex (Figs. 2a, 4a, b). The unconformity documented in the field is however intricate, deformed, and repeated by numerous thrusts (Fig. 4a). The morphology of the unconformity is studied though profile mapping of the southern-most contact between the metasediments and the igneous complex, showing that the unconformity is intensely folded (Fig. 4a). The overturned limb of the metasediments is also intensely deformed, particularly indicated by the mafic block within the metasediment unit (Fig. 4b), which shows a strong rheological contrast

between the stiff igneous complex and the weak sedimentary rocks, forming in-situ meter-scale nappe-style fold geometries (Fig. 4b). Structural and kinematic analysis of the unconformity section shows a foliation concentration dipping NNW (Fig. 4d), coherent with the foliations interpreted elsewhere as related to the regional arc-continental collision[16,17,21].

Structural studies of the autochthon along transect C-D shows penetrative shearing in the sedimentary units, with coherent structures but varying strain (Fig. 4). Typical shearing structures include asymmetric folds and/or clasts, composite-planar (S-C) fabrics, transposition of fold hinges, with strain and kinematic measurements consistent throughout the section (Figs. 4d, 7d); hence, the part of the section taken for specific quantitative kinematic analysis is assumed to represent the strain of the whole section. The marble-metapelite in Unit 3 is taken for high-resolution analysis, where the whole section is cut by a Z-shaped canyon exposing 3D structures (Fig. 7a), so the continuity and folding of units can be determined. Structural measurements for strain and kinematic analysis include foliation, lineation, fold hinges and axial planes, and S-C fabrics, illustrated in Fig. 7d.

Composite-planar (S-C) fabrics are present at all observational scales, from outcrop (Fig. 7e–g) to thin sections (Fig. 6d). The S-surfaces are defined by shape preferred orientations of micas in

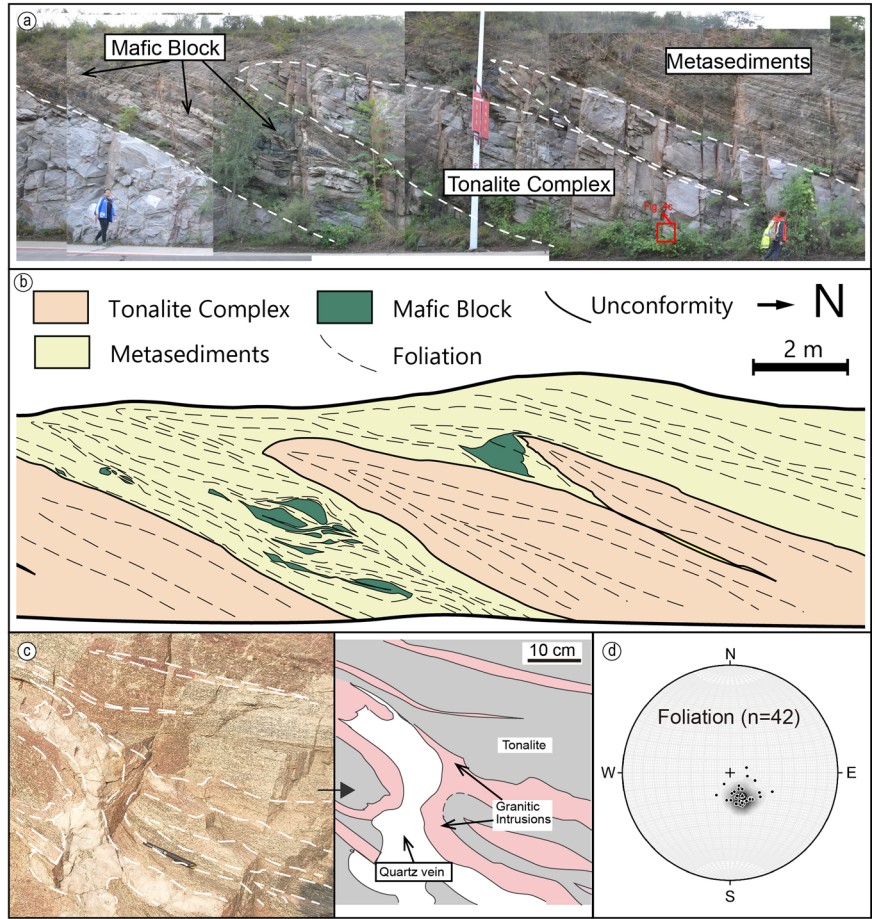

**Fig. 4 | Profile mapping of unconformity.** The location of this out crop is shown in Fig. 1d. **a** Field photos of the unconformity section. **b** Structural profile mapping, showing the structural features of the contact. **c** Photo and sketch showing the monzogranite (in pink) cross cutting the tonalite basement (in grey). **d** Kinematic data of the foliation in this section, dipping NNW.

metapelite, shape elongation and flattening of pebbles, and deformed veins, and fold-axes of isoclinal folds. C-surfaces are defined by through-going high strain zones typically demarcated by the most deformed metapelites and recrystallized marbles (Fig. 7e). Such structures are similar to the S-C-C′ fabrics described in the adjacent Zanhuang mélange and other mélanges[16,29,30]. The intersections of S-C planes can uniquely determine the sense of shear in the case of simple shear and plane strain, where the sense of shear or transport direction (slip vector) is 90° away from the intersections within the C surface on a lower-hemisphere Schmidt diagram after plotting S-C planes[29]. In progressive simple shear the finite stretching fields are asymmetric with respect to the shear plane, which would produce asymmetric isoclinal-folds and clasts as a result. Similar structures are present in the metasiltstone-metapelite from Unit 4, where the quartz veins are stretched into asymmetric boudins (Fig. 5i). This indicates that the quartz veins experienced a primary shortening but rotated into the extensional field as deformation proceeded, implying that the deformation was non-coaxial and was controlled by progressive simple shear, consistent with the regional deformation[21].

Folds are isoclinal at the meter to centimetre scale (Fig. 7f, g), and mm scales in thin section (Fig. 6d). The vergence of the isoclinal folds can be used as an indicator of the sense of shear for the overall strain field[31]. In the marble-metapelite unit, quartz veins are folded in the metapelitic matrices, where the fold axes dip northwest parallel to the foliation (Fig. 7g). From some of the rootless folds, a top-to-SE shear sense is defined. The shearing foliation is parallel to the fold axis, which suggests synchronous relationships of thrusting and folding.

Transposition of fold hinges suggests a high ductility during shearing (Fig. 6c). Fold hinges are sparsely distributed from an orientation perpendicular to lineation, to parallel, suggesting curving fold axes. The strongly curved hinge lines and a rounded, conical shape with highly curved hinges rotated into the shear direction is suggestive of sheath folds[32,33], as described from the allochthonous nappe stack above[21]. The arc distribution pattern of transport directions inferred by S-C fabrics (Fig. 7d) also supports a curving distortion of the orientation of materials during high shear strains, which coincides with the sheath fold structure. While folds with strongly curved axes can also reflect superposed folding due to multiple episodes of deformation or to progressive deformation during a single event[34], the single concentration in our kinematic data suggests a single progressive shear. Our data shows that the high intensity of the progressive deformation results in the complexity and diversity of structures.

The SSE-directed ductile shearing demonstrated in this study is consistent with previous research on the Zanhuang mélange[16,35], and is coincident with the dominant deformation episode forming the Alpine-style fold nappe/thrust belt thrust over the passive margin sequence described in this research[17,21]. Similar deformation styles of intense shearing to sheath folding is also consistent with the geometry of the Zanhuang Alpine-style nappes[17], but differs as in the autochthonous to para-autochthonous sequence described here, there are no major discontinuous shear zones (there are many minor shear zones), and the rocks show an affinity to having been derived by erosion from the basement terrane. This basement-derived provenance is further supported by the detrital zircon geochronology, described below.

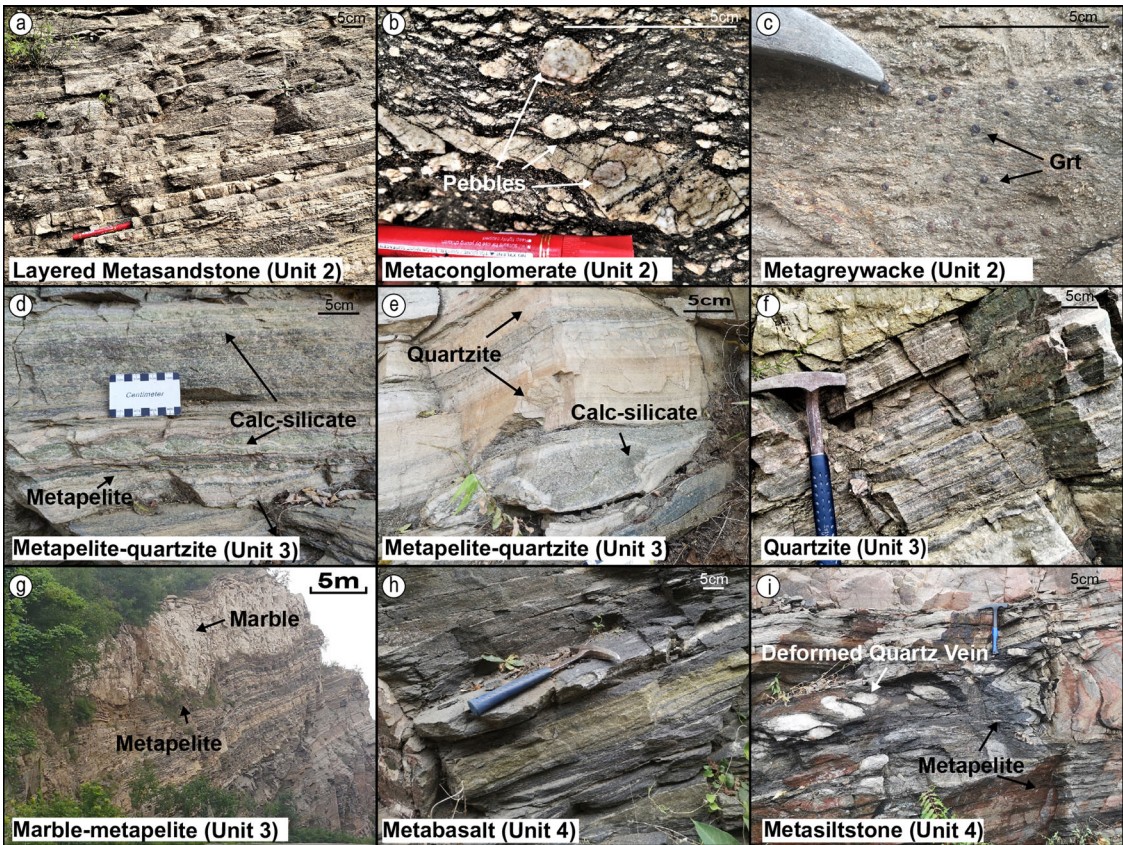

**Fig. 5 | Field photos of autochthonous metasediments. a** Sandstone with stable sedimentary banding. **b** Pebble conglomerate. **c** Greywacke with large garnet (Grt) porphyroblasts. **d** Interlayered metapelite and calc-silicate. **e** Interlayed quartzite and calc-silicate. **f** Thinly laminated quartzite. **g** Thick layer of marble and interlayed metapelite, exhibiting well-preserved bedding structure. **h, i** Intensely deformed amphibolite and metasiltstone close to the lower thrust surface of the overlying nappes.

## Geochronology constraints

Seven geochronological samples (five of metasediments, three of magmatic intrusions) were taken for detailed geochronological analysis, with results used for tectonic discrimination and age constraints. Sample 21YY-3 (37°18′35″N, 114°17′58″E) is the sandstone of Unit 2 right above the unconformity (Figs. 2a, 6a). Samples 19ZK-567 and 21YY-9 (37°19′23″N, 114°17′36″E) are taken from higher in the greywacke subunit (Fig. 2b) of Unit 2. 19ZK-567 is the metagreywacke (now garnet-bearing biotite plagioclase paragneiss; Fig. 6b), and 21YY-9 is the amphibolitic mafic dike, metamorphosed from gabbro, composed of amphibole and plagioclase, which has a clear crosscutting relationship with the metagreywacke and is deformed together with the whole sedimentary package (Fig. 2b). Sample 19ZK-024 (37°20′30″N, 114°16′47″E) is the meta-arenite of Unit 3 (now biotite quartzofeldspathic gneiss; Fig. 6c). Sample 19YY-02 (37°20′24″N, 114°15′44″E) is the metapelite taken in the marble-metapelite unit of Unit 4 (now mica schist; Fig. 6d). Sample 20NQ-1 (37°17′44″N, 114°18′47″E) is the monzogranite intrusion of the basement complex in the Xiashiliu River exposure and sample 20NQ-3 is the quartz diorite intrusion in the same outcrop (Fig. 3a).

Detrital zircons carry information about their source material, from which the depositional setting can be deciphered[36]. Owing to different provenance, sediments in convergent plate margins are usually contemporaneous with related arc magmatic products, whose age distribution should then be concentrated close to the depositional time. This is in contrast to divergent margin sediments, which typically have a larger proportion of older continental sourced zircons[37]. Based on this basic idea, we can analyze the tectonic setting from detrital zircon age distributions. The plotted age spectra in this transect show a

strong signal approaching 2510-2520 Ma (Fig. 8), suggesting active volcanos of these ages were the dominant part of its provenance. Coinciding with nappe emplacement at 2500–2520 Ma[16,17,20], the result strongly suggests that the sediments were deposited in an active volcanic zone, likely a convergent setting. Meanwhile, the detrital age distribution differs slightly in each unit. Compared to the basal sandstone (Fig. 8a), the upper sandstone in Unit 2 (Fig. 8b) consists of more old zircons and slightly more mature (more rounded) grains (Supplementary Fig. 2) implying the development of a surface drainage network that covered a broader area as the basin subsided, which facilitated the erosion, transport, and deposition of older material from older continental sources from the EB on this continental margin[8]. In contrast, Unit 3 deposits are mainly younger material from contemporaneous magmatism, with fewer older zircons (Fig. 8c, d). This transition demonstrates continent-wide control on the margin subsidence, which is strong evidence for thermal subsidence (similar to the later stages of passive margins[37]) rather than flexural subsidence (as in foreland basins[38]).

To precisely determine the depositional age and lifespan, we apply the maximum depositional age (MDA) approximation method[39]. Theoretically, the MDA is not identical to depositional age in an absolute sense, but there are cases when the sedimentary processes (<1 Ma) are shorter in duration or within the uncertainty of the analytical error (3–5% for LA-ICP-MS), making the lag between magmatism to deposition negligible[40]. For instance, in our case, the error of the measured zircon U-Pb dating age is basically 10-40 Ma, significantly larger than the normal duration of an episode of sedimentation, such that the MDA could be possibly indiscernible from the magmatic age of provenance material. Previous research suggests that this

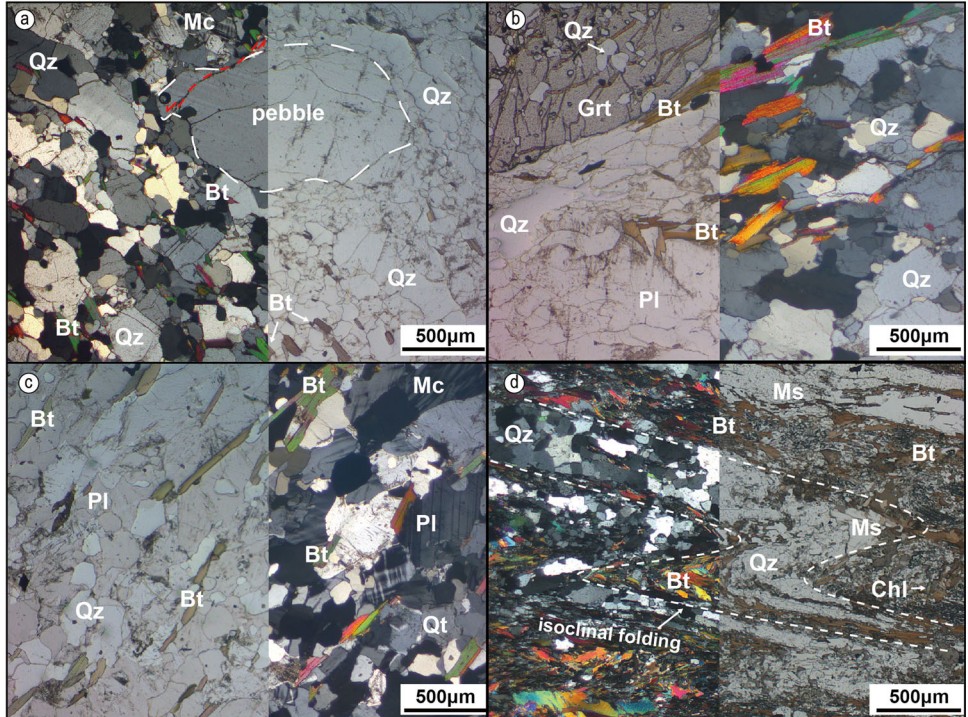

**Fig. 6 | Microscopic image of the geochronological samples. a** 21YY-3, unit 2 basal (meta)sandstone, white dash line shows the pebble within the (meta)sandstone indicating poor-sorting. **b** 19ZK-567, unit three higher greywacke. **c** 19ZK-024, unit 3 (meta)sandstone. **d** 19YY-2, unit 3 metapelite in metapelite-carbonate, showing the geometry of the isoclinal folds in micro scale on the surface vertical to foliation and parallel to lineation. Red arrow at top-right corner shows the lineation.

interpretation can only be valid if: (1) the MDA is consistent with other chronologic constraints[41–44], (2) MDAs get younger up-section[45], and/ or (3) the sediment or (meta)sedimentary rock contains first-cycle volcanic detritus[46]. The youngest single grain (YSG) method is very commonly used in sedimentary dating, but is considered untenable in most old terranes (e.g., Archaean) where Pb loss, resulting in analytical imprecision, is widespread and unavoidable[39,47]. We therefore apply four additional calculations: YC1σ (youngest single cluster overlapping at 1σ uncertainty), YC2σ (youngest single cluster overlapping at 2σ uncertainty), YPP (youngest probability peak in mode of kernel density estimation)[48], YSP (youngest statistical population)[39], whose calculation are explained in Supplementary Text.

All calculations produce results with decreasing ages up section within error, yet YSG and YC1σ in 19YY-02 are distinctly younger, for which these two methods are discarded. On the other hand, the τ method (same as YPP in this research), YSP and YC2σ, tested to have >98% reliability to be no younger than true depositional age[39], is favoured and applied in the following section. The MDA of the base of the section gives an ambiguous result with a few overlapping estimates varying from 2563 Ma (the oldest possibility of YC2σ) to 2528 Ma (YPP age, the youngest among all calculations), while the uppermost sample yields a constraint from 2534 Ma (the oldest possibility of YSP) to 2504 Ma (the youngest possibility of YC2σ). Our result shows that all calculations are within the youngest peak range, and among all, the conservative YC2σ calculation is chosen for its consistency with geological relationships, and an estimated > 98 % reliability[39]. In conclusion, the sequence yields a 2558 ± 5 Ma subsidence/sedimentation initiation age, and a 2510 ± 6 Ma sedimentation termination age.

Other chronological constraints based on the principle of cross-cutting relationship are also applied to verify the MDA approximation. We analysed the age of a cross cutting mafic dyke mapped within the greywacke of Unit 2 (Fig. 2b), which yields a U-Pb age of 2507 ± 11 Ma (Fig. 5a), showing good consistency with the geological relationships in the field and our conclusions on the best ages for our sequence. Our result shows (1) good consistency with field relationships and regional framework, including the initial ages of the sequence (2558 Ma) are close to previously estimated ca. 2550 Ma basement formation[8], and the termination ages (2510 Ma) are older than nappe emplacement at ca. 2500–2520 Ma[17,20](Supplementary Fig. 1a); (2) decreasing age trend up-section within errors and concentration in specific ranges (Fig. 2); and (3) dominant composition of first-cycle volcanic detritus, by zircon of strong oscillatory bands in the cores, rimed with narrow metamorphic rim domains indicating igneous origins, and by the sub-angular and rather complete zircon morphology indicating a short transportation duration (Supplementary Fig. 2). Therefore, the MDA can be considered as a close representation of the true depositional age in our study, and the sequence can be constrained to c. 2558 Ma–c. 2510 Ma, with ~50 Myr lifespan.

## Discussion

The Zanhuang carbonate-siliciclastic package lithologically resembles typical passive margin sequences[26], with platformal carbonate overlying coastal sediments upon a felsic continental basement. However, our new geochronological data, together with regional syntheses showing that the autochthonous shelf sits in an accretionary orogen[8,15,49], is not consistent with interpretation of the sequence as a "rift-to-drift" passive margin of a classical Wilson Cycle. Here we discuss a viable alternative for the passive margin sequence.

In terms of regional tectonic setting, the basement to the EB consists of a series of arcs and/or microcontinents, accreted from 2.7–2.55 Ga[8,14,15], to form a relatively stable block, likely thickened by multiple collisions by 2.58 Ga (the age of the base of the autochthonous sequence), with the Wutai/Fuping arc accreted by 2.50 Ga. This collage was then affected by an additional series of Paleoproterozoic collisions, adding the various components of the WB (Fig. 1a). Thus, the NCC is an Archaean-Proterozoic accretionary stage orogen[1], and not a typical rifted older continent blanketed by an Atlantic-type passive margin. The age of the ocean outboard of the Zanhuang

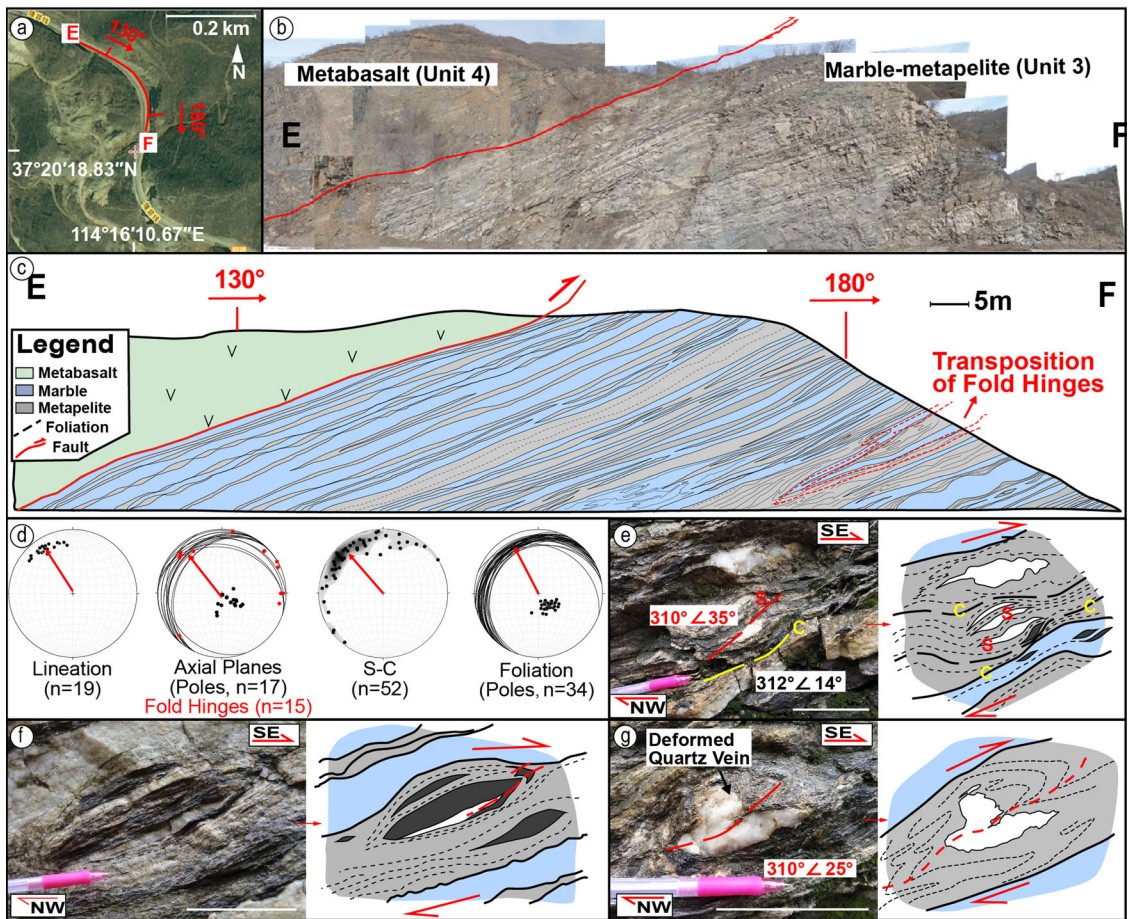

**Fig. 7 | High-resolution structural profile, measurements, and sketches of the marble-metapelite unit (Unit 3). a** Transect satellite imagine and GPS from Map World (天地图, www.tianditu.gov.cn), National Platform for Common Geospatial Information Service. The location is shown in Fig. 1d. **b** Section pictures taken by UAV drones and merged using Adobe Photoshop. **c** Large-scale 3D digital structural mapping (scale, 1:100). **d** Measurements of the outcrops, with concentration dipping to NW, and the curving distribution of the S-C fabric. **e**–**g** Photos (left) and sketches (right) of kinematic index structures, showing penetrative SSE shearing. White bar in the photos indicates 5 cm.

carbonate-siliciclastic sequence, shown to be >2698 Ma by the age of volcanics related to subduction initiation in the overlying allochthonous nappe[17], is significantly older than the age of initiation of the passive margin-like sequence (ca. 2558 Ma), which is also incompatible with formation as a classical Wilson cycle passive margin, but compatible with formation in a complex of active and previously accreted arcs in the EB at ca. 2.55 Ga[8,14,15]. Thus, the results strongly suggest the autochthonous passive margin sequence was deposited concurrently with accretion of multiple arcs, with intervening oceans of different ages, in an accretionary orogen, similar in style to the Altaids of central Asia[50,51].

We propose that the passive-margin-like sequence was generated by subsidence of the backside of a previously accreted arc: the amalgamating arcs of the EB formed a protocontinent, whose trailing margins subsided, but unlike most modern passive margins that experience thermal subsidence following rifting, the Zanhuang margin subsided after accretion and cessation of volcanism, generating similar thermal-subsidence-related platformal or continental shelf-like environments, and remaining temporarily stable before the next arc collided and joined the amalgam 50 Myr later (Fig. 9).

We further test two hypotheses for subsidence. The first is that the basin is a flexural-controlled foredeep related to the approaching fold-thrust belt[6,52]. This interpretation is rejected because the deposition initiation age is older than the age of initial nappe encroachment at ca. 2510-2520 Ma[17]. The second is thermal subsidence along the backside of the archipelago after its collision with older previously accreted arcs

forming the growing Eastern Block (Fig. 9). The collision would cause a cessation in magmatism and thermal subsidence in the structurally thickened growing protocontinent. Post-collisional back-arc thermal subsidence basins such as the Zanhuang carbonate-siliciclastic sequence record initial stabilization of amalgamated segments of juvenile continents in accretionary orogens. Possible modern analogues include the Kermadec arc-system, which has developed a gentle platform on its backside[53], with ongoing subduction and accretion in the forearc, and meanwhile accumulating muds with high carbonate and silica contents[54]. This consistency with the North China Archaean example possibly indicates a similar mechanism of post-collisional thermal subsidence in accretionary stage orogens, marking a fundamental stage in the formation and stabilization of continents. This novel tectonic setting for subsidence on the quiet backside of accreted arcs is a previously overlooked signal of protocontinental growth and stability, and may be more common in continents formed by amalgamation of archipelagos than presently appreciated.

The formation of the Zanhuang and other related passive-margin-like sequences in the North China Precambrian orogenic system occurred at a critical interval of Earth evolution, with the emergence of large continental landmasses[1,2], coincident with initial stages of the Great Oxidation Event (Fig. 10), reflecting a major transition in life and the environment on the planet[3]. The first consequence of gentle thermal subsidence of a continental margin is the formation of a shallow water carbonate platform. Whilst some carbonates form through purely chemical reactions in redox environments, these

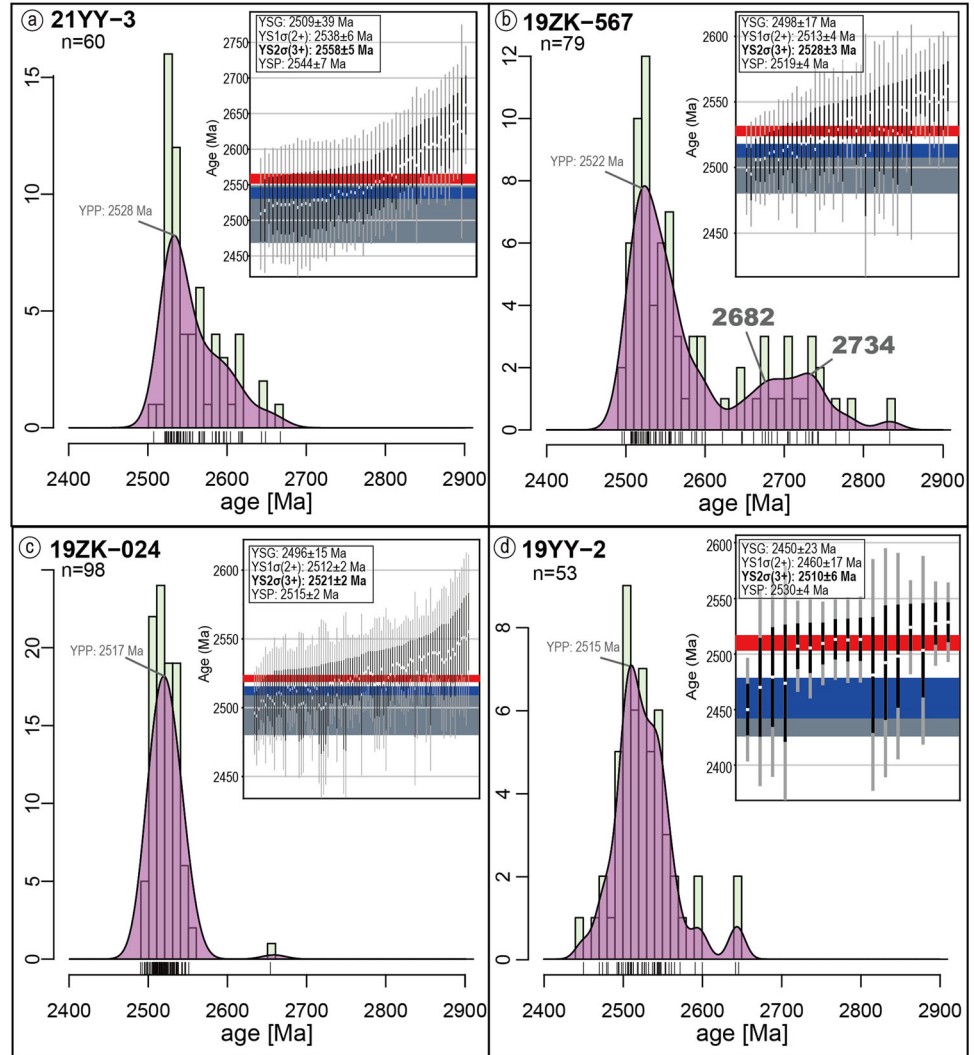

**Fig. 8 | Detrital ages spectrum diagram and MDAs.** For Abbreviations refer to Fig. 2. YPP is labelled in the spectrum, and YSG, YC1σ, YC2σ, YSP results are shown in the inset. **a** 21YY-3, metasandstone at the base of Unit 2. **b** 19ZK-567, greywacke at the top of Unit 2. **c** 19ZK-024, quartzofeldspathic arenite in Unit 3. **d** 19YY-2, metapelite at the top of Unit 2, interlayered with carbonate. The inset shows the distribution of the chosen cluster, where the white points show the analysed results. For the vertical bars, black/light grey indicates 1σ/2σ error; for the horizontal bar, dark grey indicates YSG, blue indicates YC1σ, and red indicates YC2σ, with errors. Data of this figure can be referred to Supplementary Dataset 2.

typically precipitate iron-rich deposits (banded iron formation);[55] the Zanhuang carbonates are not associated with iron formations, implying a biogenic origin. Other studies suggests that bioactivity as recorded by stromatolites led to thick accumulations of Archaean carbonates[56–58].

The Zanhuang carbonate (Unit 3) platform formed in an arc-accretionary tectonic setting within a 50 million year interval (Fig. 8c, d). Biogenic shallow water carbonate sequences formed over 10's–100's Myr time intervals are preserved in several Archaean terranes in both shallow marine (continental margin) and non-marine (terrestrial lake) settings (Fig. 10). The former includes the Strelley Pool Formation of the Warrawoona Group in the Pilbara Craton (ca. 3.5–3.4 Ga)[56], the Chobeni Formation in the Nsuze Group of the Pongola Supergroup (ca. 3.0–2.9 Ga)[57,59,60] and the Campbellrand-Malmani carbonate platform (ca. 2.58–2.52 Ga)[58,61] in the Kaapvaal Craton, the Mosher Carbonate of the Steep Rock Group in the Superior Province (ca. 2.82–2.78 Ga)[62] and, as reported here, the Zanhuang carbonate platform (ca. 2.52–2.51 Ga) in the North China Craton. The latter group include intra-continental or intra-crater lake carbonates of the Tumbiana formation of the Fortescue

Group (ca. 2.74–2.72 Ga)[63] in Pilbara Craton, Hartbeesfontein Basin of the Ventersdorp Supergroup (ca. 2.78–2.71 Ga)[64] in the Kaapvaal craton, and within the Joutel Volcanic Complex in the Abitibi Sub-province (ca. 2.73–2.72 Ga)[65] in the Superior Province of the North American craton. The presence of biogenic carbonate suggests that Archaean Cyanobacteria may have been more prolific than currently appreciated[66], and promoted the accumulation of free oxygen in Earth's atmosphere – from regional oxygen oases (or whiffs) to the Great Oxidation Event[3,59,60,62,67,68]. In the transgressive continental margin setting on the basement tonalitic complex, the shallow marine platforms would receive weathering products that would initially supply nutrients for biological growth[59,60,62,68]. Compared to terrestrial lake, marginal carbonate platforms generally occupy larger areas with a wider range of paleoenvironments, and could have more potential global impact. Therefore, as protocontinents were stabilized after accretion of arc terranes or archipelagos, subsidence of continental margins on amalgamated arc back-sides would contribute to the formation of initial shelf habitats for microorganisms[52,60], that generate regional oxygen oases in an anoxic atomsphere[60,61,66].

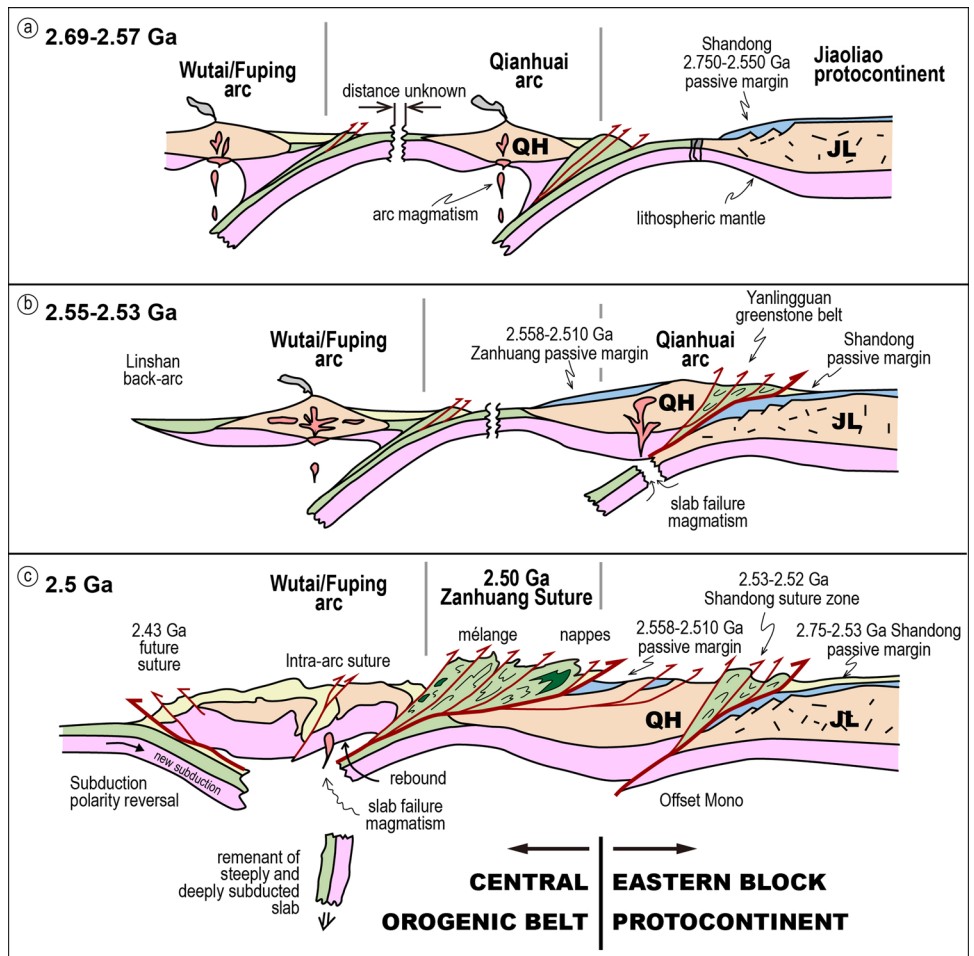

**Fig. 9 | Schematic of regional tectonic evolution. a** EB protocontinent formed by arc amalgamation before ca. 2.55 Ga. **b** From ca.2.55 Ga the margin subsided forming a passive margin. **c** At ca. 2.51 Ga, arc-continent collision caused nappe emplacement[17] and termination of passive margin sedimentation.

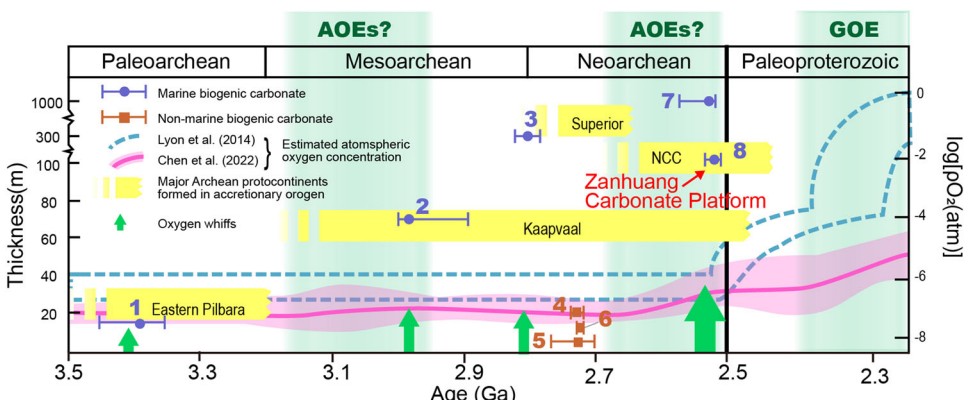

**Fig. 10 | Schematic diagram illustrating connection between protocontinent emergence in the Archaean and biogenic carbonate distribution.** *Violet/bronze* solid point with error bar for: depositional age of marine/non-marine biogenic carbonate. *Blue curve*: emerging model of oxygen level ($\rho O_2$(atm), atmospheric partial pressure of $O_2$)[68]. *Pink Curve*: atmospheric $O_2$ variation using machine learning[67]. *Green arrows*: possible 'whiffs' of $O_2$ coupling carbonate platform emergence. *Vertical green bars*: suggested oxidation evens, from[67]. GOE: Global oxidation event. AOE: Archaean oxidation event. (1) Strelley Pool Formation, Pilbara Craton (ca. 3.5–3.4 Ga)[56]. (2) Chobeni Formation, Kaapvaal Craton (ca. 3.0–2.9 Ga)[57,59,60]. (3) Mosher Carbonate, Superior Province (ca. 2.82–2.78 Ga)[62]. (4) Tumbiana formation, Pilbara Craton (ca. 2.74–2.72 Ga)[63]. (5) Hartbeesfontein Basin, Kaapvaal craton (ca. 2.78–2.71 Ga)[64]. (6) Stromatolite in Joutel Volcanic Complex, Superior Province (ca. 2.73–2.72 Ga)[65]. (7) Campbellrand-Malmani carbonate platform, Kaapvaal Craton (ca. 2.58–2.52 Ga)[58,61]. (8) (this study) Zanhuang carbonate platform, North China Craton (ca. 2.52–2.51 Ga).

In the Paleoarchaean and the Mesoarchaean, stabilization of individual protocontinents, particularly the Pilbara[56] and Kaapvaal Cratons[57,60], was followed by the growth of marginal carbonate platforms possibly resulting in shallow water oxygenation[58]. In the Neoarchaean (2.8–2.5 Ga), the increasing amount of landmass leads to augmenting volume of biogenic marginal carbonate sedimentation globally, in the Superior, Kaapvaal, and North China Cratons. While the Zanhuang example is only one case, its well-documented link with the formation of the first proto-continental nuclei of the craton, at a time when many cratons were amalgamating to form stable continents and emerging[1–3], indicates that such a process played a pivotal role in promoting the atmospheric oxygenation in the early Earth. A recent machine learning study suggests the onset of regional oxygenation events in the Archaean, followed by global oxygenation shortly after the Archaean-Proterozoic Boundary[67]. The sudden outburst of biogenic carbonate platforms globally around 2.5 Ga (Fig. 10) may thus share a common triggering mechanism, showing a link of deep Earth processes such as a transition from bottom-up to top-down mantle convection[69] and the oxygenation of the atmosphere near the Archaean-Proterozoic boundary[3,66–68]. As protocontinents were generally stabilized by intraoceanic arc amalgamation around the Archaean-Proterozoic boundary[2], biogenic carbonate platforms were built as a consequence. The sudden rise of oxygen from these oases eventually led to global change and the later GOE event started about 2.45 Gyr, making Earth a more habitable planet[66–69].

## Methods

### Tectonostratigraphic mapping and structural analysis

Field mapping and profile construction was performed at a scale of 1:100 with no vertical or horizontal exaggeration. All structural elements (e.g., foliation, lineation, fold axis, axis plane, thrust fault) were carefully collected, measured in 3D exposed surfaces and depicted in lower hemisphere equal angle projections. The program Stereonet 11 by Rick Allmendinger was used to create these plots.

### Zircon U-Pb dating

In situ U-Th-Pb geochronology were conducted by laser-ablation-inductively-coupled-plasma mass-spectrometry (LA-ICP-MS) in the GeoHistory Facility of the John de Laeter Centre (JdLC), Curtin University, Perth, Western Australia, Australia and the Wuhan SampleSolution Analytical Technology Co., Ltd., and the State Key Laboratory of Geological Processes and Mineral Resources, China University of Geosciences, Wuhan, China. Analytical details see Supplementary Discussion. Zircon U-Pb data is visualized by software IsotopR and the MDA calculations (expect the YSP by Excel) are by software *detritalpy*.

## Data availability

The geological maps, structural data, detailed description and location of the geological samples are provided in the main text. Remaining parts of samples are stored permanently in the sample storage facility in the Center for Global Tectonics at China University of Geosciences, Wuhan. The processed geochronological data and zircon CL images are provided in Supplementary Data/Information.

## Code availability

Software and codes used for this study are available in published works[43,70] as noted above.

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

## Acknowledgements

This work is supported by the National Natural Science Foundation of China (Grant Numbers: 41890834, 41888101, 41961144020, and 42072228), Chinese Ministry of Education (BP0719022), and the MOST Special Fund (MSF-GPMR2022-7) of the State Key Laboratory of Geological Processes and Mineral Resources, China University of Geosciences, Wuhan. Research in the GeoHistory Facility, JdLC is supported by AuScope (auscope.org.au) and NCRIS. We thank M.Z. and X.W. for field assistance.

## Author contributions

T.K. and L.W. conceived the project and obtained funding and supervised this work. The manuscript was initially drafted by Y.P., and was revised by T. K., L. W., and N.J.E.. Y.P., T.K., L.W., Z.L., C.W., X.L., and Y.Z. participated in the fieldwork and analysis; Y.P., Z.L., C.W., X.L., Y.Z., and N.J.E. participated in the laboratory work and data analysis.

## Competing interests

The authors declare no competing interests.
