## [Peer Review File · Nature Communications]

Passive margins in accreting Archaean archipelagos
signal continental stability promoting early atmospheric
oxygen riseEditorial Note: Parts of this peer review file have been redacted as indicated to avoid any copy right infringement.

Reviewer #1 (Remarks to the Author):

Thanks for the opportunity to review this very interesting manuscript, the authors present good field observations and nice data on the Archean North China Craton and the Great Oxidation Event with the global vision. The manuscript should be of general interest to a wide audience. It is well written and the figures are all extremely well-done and convey the message. I recommend publication after minor revision, with a few suggestions below, my concerns mainly include: (1) this manuscript showed a nice field observation in the Zanhuang mélangé of the North China craton, but I would suggest the authors should improve the field interpretations a little bit, such as Figure 4, the authors might miss some important information among the blocks, and also some different foliations/boundaries; (2) I didn't see the carbonate in the field photos, maybe, the authors forget to identify them, however, it's such a really nice parametamorphic rock association in the Neoarchean Zanhuang mélangé; (3) In the Figure 1D, they should be foliations? but they were identified as like the bedding. Scales are need in the geological maps and all filed photos. I don't comment on the tectonic model discussion, it looks reasonable to me.

Reviewer #2 (Remarks to the Author):

NCOMMS-22-29586

Emergence of passive margins in accreting Archean archipelagos signals continental stability and creation of pre-GOE oxygen oases
Peng and Kusky et al.

Review by Ross N. Mitchell (ross.mitchell@mail.iggcas.ac.cn)

This is an excellent idea with field data to support it about Archean geology with implications for the leadup to the GOE. The accretion of ocean arc archipelagos is likely how many cratons formed—explaining nicely the granite–greenstone architecture of most Archean cratons. The authors present the clever idea that the passive margins of Archean archipelagos that accreted to form protocontinents would have, upon continental stabilization, subsided and created shallow water ecospace in which photosynthetic bacteria could have thrived and therefore eventually caused the tipping point of the GOE. Although their locality, North China craton, is just one, the authors are correct to point out that it is the *cumulative* effect of the cratons stabilizing globally that would have led to this increased ecospace over time, with later-forming cratons like North China being the proverbial “straw that broke the camel’s back”. The work is detailed and robust and the idea is compelling and fitting for publication in a general journal like Nature Communications. Although my comments below may seem critical, they are made only in an effort to improve an already strong and provocative paper. I would recommend MINOR REVISIONS.

Major comments

(1) The pre-GOE oases, or pre-GOE “whiffs” of oxygen should not be taken as fact and the authors must admit somewhere that there is uncertainty about their existence. In fact, they are quite contentious and potentially dubious post depositional diagenetic artifacts (Slotznick et al., 2022). There is also depositional modeling inspired by the Archean presence of redox-sensitive detrital grains of pyrite and uraninite (easily weatherable if free oxygen is available anywhere in the atmosphere) that casts theoretical doubt on the existence of pre-GOE “whiffs” of oxygen, suggesting the rise of oxygen must be younger than 2415 Ma (Johnson et al., 2014). I would therefore recommend the authors instead focus on the “leadup” or the “prelude” to the GOE. 2.5 Ga is not the start of the GOE. Some think the GOE was late as 2.33 Ga (Luo et al., 2016). Even if as old as <2415 Ma (Gumsley et al., 2017), the authors’ archipelago subsiding backsides are still not a “coincidence” with the GOE, but a prelude or leadup to it. Perhaps the delay can be explained in terms of subsidence creating sufficient

accommodation space that shallow platformal shelves covered enough areal extent to finally promote the full-fledged GOE. Furthermore, the more arcs that protocontinents accreted, the more continents would stabilize and such larger and higher continents would shed more sediment, making it more likely to form broad continental shelves to provide the ideal shallow water ecospace for photosynthetic bacteria. Also, it could be mentioned that sourcing rock-derived phosphorous as nutrients (which would be rich in apatite in basalts of ocean arcs) is a prerequisite for the GOE (Cox et al., 2018). Whereas evidence of photosynthesis may substantially pre-date the GOE (Nutman et al., 2016), nutrients were likely the rate-limiting reactant, which the authors' model can nicely explain.

(2) In the profile description, it seems the island arc (graywacke) and backside sediments (calc-silicate) are conformably contacted? If so, why do the carbonates necessarily occur in a backside setting? These sediments can simply be part of the arc (arcs may have hiatus). Some Phanerozoic intra-oceanic island arcs have volcanoclastic rocks interbedded with the carbonate. If the island arc is in fault contact with the backside sediments, are these backside sediments comparable with the continental passive margin sediments? If they are, it means the island arc is in fault contact with both sides of the sediments. Then the simplest way to understand the arc-continent collision process is that the arc might have been thrust onto the continent (like the Macquarie Arc in SE Australia).

(3) Arc-continent collision process might be the direct contribution to any variations of the environment in this case (?). You have constantly new passive margins replacing the old ones. There is nearly no net increase of passive margin only from this process (assuming the length isn't changing as it's not discussed here). The continent is growing, but the passive margins are just constantly being replaced. The direct observations from the evidence shown in this paper are 1) arc-continent collision processes resulted in 2) continental growth/addition of carbonate. Thus, it would be clearer to state that arc-continent collision process is the key (?). The proposal of Phanerozoic arc-continent collisions in the tropics set Earth's climate state might be worth to be referred (Macdonald, Francis A., et al. Science 364.6436 (2019)).

Minor comments

By my count the current title is two words over the 15-word NC limit. I might suggest: Passive margins in accreting Archean archipelagos signals continental stability and creation of pre-GOE oxygen oases

(I would avoid the word "emergence" for meaning "appearance of" or "rise of" passive margins given that the paper also discusses continental emergence, i.e., above sea level.)

Archaean should use UK spelling for this journal

Abstract is quite good, but could be better. Some words to delete are suggested that may allow for a better re-write.

Line 19 see comment about not using "emergence" unless continental emergence

Line 19 late Archean should not be hyphenated

Line 22 passive-margin type

Line 26 awkward wording "open-ocean back-side"?

Lines 27 and 28 shallow water

Line 28 photosynthetic microbes

Line 29 still don't know what archipelago "back-sides" are? Also the backside (of a hill, for example) is an unhyphenated word.

Line 29 represents

Line 29 delete “environmental”; it is either redundant or even wrong (niches are ecological phenomena)

Line 31 Aren’t “initial” and “proto”(continental) kind of redundant?

Line 32 Archaean–Prot boundary most certainly isn’t “early Earth”, which is typically applied to Hadean and the earliest Archean eras only. Delete and save some words for better abstract.

Line 34-37 Quite a word salad for a first sentence! Do you really need to lead with the contentious onset of PT? Largely irrelevant by late Neoproterozoic—when likely everyone in the world, with few or one exception, already expects PT would have been operational. Why not just jump into the contentious GOE and the even more contentious pre-GOE “whiffs” of oxygen

Line 38 submerged

Lines 34-40 short first paragraph (especially if you delete the essentially irrelevant onset of PT debate; previous comment); consider moving Lines 41-45 up to paragraph one.

Line 45 “In this study, we report...” (NC formatting guidelines)

Line 45 passive-margin type shallow water sequence (these hyphenation corrections will not be repeated but should be corrected throughout)

Line 46 northern China (North China not a formal term)

Line 48 fuzzy is too colloquial for scientific writing; synonym uncertain/incomplete/etc.

Line 49 Possessive with inanimate nouns is too colloquial for scientific writing; rewrite

Line 52 nearly 100 or even 200 Myr in between the A–P boundary and the GOE....

Line 53 Results section should start here. Starting with a sub-section “Geologic setting.”

Line 56 lowercase citations of figure panels in NC (fix throughout)

Line 64 Southwest

Line 66 use (refs. X) to avoid confusion with units (fix throughout when an ambiguity)

Line 77 for numeric ranges, en dash (–) should be used and without spaces before/after (fix throughout)

Line 278 upsection... and maybe listed the older age first to make more intuitive the upsection younging trend

Line 280 transport

Line 281 on this continental margin (since deposition is last in sequence)

Line 287 define abbreviation

Line 293 than the normal duration of an episode of sedimentation

Line 292 such that the MDC

Line 303 in the Supp...

Line 310 The MDA

Line 321 verify/validate/test (validate not a word)

Line 325 unclear meaning of "above".. overlying sequence or aforementioned sequence?

Line 334 close representation of the true depositional age (also no need to introduce an abbreviation used only once.. and on the *previous* page!)

Line 354 avoid zombie compound nouns...the volcanics related to subduction initiation

Line 357 Wilson cycle

Line 362 sometimes fully hyphenated, sometimes not; be consistent

Line 366 thermal-subsidence-related

Line 368, 401 50 Myr later (duration not age)

Line 436 Again, see Major Comment about timing of authors' mechanism and GOE

Figs can be better.

Fig 1. keep the font consistent.

Fig. 1A. the degree sign is missing (e.g., 100° E); the red square (for B) can be more obvious; keep the cross-section lines format consistent (a-b, c-d both with/without bars at the ends)

Other figs: keep the upper/lower cases consistent for the units. For example, both CM and m were used; Line thickness of the legends; Keep the north arrows consistent; Use a big arrow for a map (like in Figs 1 and 3); For a cross-section, an arrow with a tail is more often used like Fig 7 better than Fig 4

Cox, G. M., Lyons, T. W., Mitchell, R. N., Hasterok, D., and Gard, M., 2018, Linking the rise of atmospheric oxygen to growth in the continental phosphorus inventory: *Earth and Planetary Science Letters*, v. 489, p. 28-36.

Gumsley, A. P., Chamberlain, K. R., Bleeker, W., Soderlund, U., de Kock, M. O., Larsson, E. R., and Bekker, A., 2017, Timing and tempo of the Great Oxidation Event: *Proceedings of the National Academy of Sciences*, v. 114, p. 1811-1816.

Johnson, J. E., Gerpheide, A., Lamb, M. P., and Fischer, W. W., 2014, O₂ constraints from Paleoproterozoic detrital pyrite and uraninite: *Geological Society of America Bulletin*, v. 126, p. 813-830.

Luo, G., Ono, S., Beukes, N. J., Wang, D. T., Xie, S., and Summons, R. E., 2016, Rapid oxygenation of Earth's atmosphere 2.33 billion years ago: *Science Advances*, v. 2, p. e1600134.

Nutman, A. P., Bennett, V. C., Friend, C. R. L., van Kranendonk, M. J., and Chivas, A. R., 2016, Rapid emergence of life shown by discovery of 3,700-million-year-old microbial structures: *Nature*, v. 537, p. 535-538.

Slotznick, S. P., Johnson, J. E., Rasmussen, B., Raub, T. D., Webb, S. M., Zi, J.-W., Kirschvink, J. L., and Fischer, W. W., 2022, Reexamination of 2.5-Ga "whiff" of oxygen interval points to anoxic ocean before GOE: *Science Advances*, v. 8, no. 1, p. eabj7190.

Reviewer #3 (Remarks to the Author):

This paper reports a suspicious Neoproterozoic passive margin-type sequence from the North China craton. Tectonostratigraphic and detrital zircon analyses are approached to decipher its nature. It is concluded that this unit was sitting on the open-oceanic back-side of an amalgamating arc terrane in a shallow-water environment due to local thermal subsidence. If this is approved, it represents a stable environment that is distinct from the typical passive margins, but might be a specific environment signaling initial protocontinental maturity. This environment might provide a new opportunity for oxygenic life prior to the Great Oxidation Event. This study is innovative - not only declaring a unique sequence at the initiation of continental maturity in the Neoproterozoic, but also explaining the possible mechanism of subsidence of early continent to create shallow-water environment to be capable of early oxygenic life.

My major concern is the age of the key tectonostratigraphy in the paper. Although it has not been clarified directly in the paper (it should have had), it is quite clear that the key sequence includes the Gantaohu Group based on the geological map (Figure 1) and the description of its four units. There are many previous local and international publications for this group (e.g., Liu et al., 2012; Du et al., 2016). It is well-known that it is the Paleoproterozoic in age since its definition in 1960s (for example, there is an age of ~2.1 Ga for volcanics: Du et al., 2016). All these previous data on the group should be at least reorganized and evaluated.

Another major concern is the interpretation on the tectonic environment of the sequence. It is quite depending on the author's own model on the tectonic evolution of the region, as well as their interpretation on the age of the Gantaohu Group (see above). Based on the sedimentary facies of the group, as well as the chemical and geochronological data already reported, it has been proposed that the Gantaohu Group was developed on the continental crust, either in an intra-continental rifting environment (e.g., Du et al., 2016 and many other researches) or a back-continental arc environment (e.g., Liu et al., 2012).

In addition, both the mechanism for the subsidence of the so-called 'Neoproterozoic' passive marginal basin and its link with the Great Oxidation Event are not adequately supported by the data provided in the paper (hardly proved but just stated). Is there any sign of thermal subsidence of the region? Is there any fossil or tracer to prove the nourish of oxygenic life in the sequence?

With the above concerns in mind, I think this innovative work is rather inadequate at this stage. I would be happy to see more evidence to depict it – if it has been proved, it would have provided a totally new concept to the research field, which with no doubt would be revolutionary.

Du, L., Yang, C., Wyman, D. A., Nutman, A. P., Lu, Z., Song, H., Zhao, L., Geng, Y., and Ren, L., 2016, Age and depositional setting of the Paleoproterozoic Gantaohu Group in Zhanhuang Complex: Constraints from zircon U–Pb ages and Hf isotopes of sandstones and dacite: *Precambrian Research*, v. 286, p. 59-100.

Liu, C., Zhao, G., Liu, F., Sun, M., Zhang, J., and Yin, C., 2012, Zircons U-Pb and Lu-Hf isotopic and whole-rock geochemical constraints on the Gantaohu Group in the Zhanhuang Complex: Implications for the tectonic evolution of the Trans-North China Orogen: *Lithos*, v. 146-147, p. 80-92.

Abstract:

Line 19-21: this statement on the significance of 'the emergence of stable shallow-water continental platforms' is rather arbitrary.

Line 26: how do you know it was on 'the open-ocean back-side'? If it was on the open-ocean back-side, we won't call it a 'passive margin'?

Line 28: it is likely that there was no indication of 'microbes' in the stratigraphy (e.g., there was even no stromatolite in the carbonates).

Line 48: 'this data' should read 'these data' or 'the dataset'.

Line 53-59: many geologists suggested that the eastern craton and western craton were collided at 1.9-1.8 Ga, rather than 2.9-2.5 Ga. Please at least mention this option.

Line 60-63: '... causing deformation and metamorphism between 2.7-2.55 Ga to form protocontinental nuclei...' – however, there are hardly metamorphism ages of 2.7-2.55 Ga in the North China craton. Actually, the widespread metamorphism in this craton was happened most likely after 2.55 Ga. BTW, 'nucleii' should read 'nuclei'?

Line 72-80: the regional geology should start with the units defined by local workers, and then you may specify your tectonic interpretation for each unit.

Line 83-90: I see no evidence of allochthonous bodies/units in the Zhanhuang region. Based on what you define them as 'allochthonous' units? Ages different from the very small local basement? The tectonic contact among units? No strong speculation has been stated here or hereafter. The whole speculation is suspicious.

Line 102-104: 'The quartz diorite ... characteristic of mid-crust intrusions'. I do not aware that quartz diorite intrusives should be mid-crustal. Please show the observations and evidence.

Line 106-108: which episodes of deformation? Late Archean or Paleoproterozoic (episodes as revealed by both zircon U-Pb and amphibole Ar-Ar ages)?

Line 109: 'cover'? Which is/are the 'cover'? Please specify.

Line 111-144: what are the thickness of each unit (without counting the deformation)?

Line 121-124: all these sequences were metamorphosed into high-amphibolite to granulite facies without any preservations of grading. How do you know they are graywackes rather than shales to siltstones?

Line 131: 'shallow-water mudstone'? Based on what?

Line 134-135: 'low energy environment'?

Line 136-138: The arenite comprised by quartz, feldspar and biotite has 'minor' compositional differences with quartzite??

Line 143: how do you know the 'marbles' are originally 'limestone'?

Line 146-151: why this metabasalt-featured unit allochthonous slice rather than a part of the stratigraphy? The whole sequence was well-known as the Gantaohu Group in the literature. Although it has undergone stronger deformation in the middle of the Zhanhuang region, it shows quite consistent sequences with those west to the region will less deformation. It is well-known to be a Paleoproterozoic intracontinental sequence.

Line 149: It should be noticed that the Archean basement in the COB has varied ages from 2700-2500 Ma.

Line 152-155: The statement here is arbitrary.

Line 157-159: Observation is needed to show an overthrust allochthonous unit. A tectonic/fault contact does not guarantee an allochthonous origin.

Lines 165-166: Sure thing. The basement is older than 2.5 Ga; while the strata was middle Paleoproterozoic in age.

Line 169-176: yes, there was probably an episode of metamorphism at the late Archean, but the major episode was the late Paleoproterozoic.

Line 182-186: Again, it is reasonable that the deformation of the basement and the Gantaohu Group (unit 1-4) is different.

Line 190-247: sorry I did not read it in quite detail. It is really hard to follow as the major deformation here, or at least the deformation as revealed by Ar-Ar ages is the late Paleoproterozoic. Without distinguishing the Archean and Paleoproterozoic episodes of deformation, it is really hard to evaluate the description.

Line 266-269: the error of these ages are quite large, over 20 Ma or 30 Ma. There are no obvious differences between these age groups to discriminate their provenances.

Line 270-286: I admit that there could be a successive thermal/igneous event in the region during the Late Archean; then how can you distinguish ages between domains with lead loss and domains not, especially considering their large errors of singular ages.

Line 287-335: all these analyses and interpretation is based on the assumption that the sequence was the late Archean in (depositional) age. But that was not the case, the volcanics (metabasalts) and other rocks show clear evidence that the rocks are middle Paleoproterozoic in age (e.g., Du et al., 2016; Liu et al., 2012).

Line 321-325: it analyzed the age of a crosscutting metamorphosed mafic dyke, which gives a U-Pb zircon age of 2507 Ma. I highly doubts whether this is an age of crystallization or an age from inherited zircon grains, and hence, it is inadequately proved that this ~2507 Ma age is a key to constrain the deposition age of the tectonostratigraphy.

Line 338-340: But the whole basement was not exhumed until the Late Paleoproterozoic; how can it be the exposed coast in the Archean.

Lines 358-361: 'Thus, the results strongly suggest the autochthonous passive margin sequence was deposited concurrently with accretion of multiple arcs, with intervening oceans of different ages, in an accretionary orogen, similar in style to the Altaids of central Asia'. All the ages provided in the paper are within-error quite similar, and nothing of the kind can be supported.

Line 374-376: what is the evidence for 'thermal subsidence' of the region in the Late Archean?

Line 436-437: 'The sudden out-burst of biogenic carbonate platforms globally around 2.5 Ga ...'. Where does this statement come from? I would not show the dataset here, but the 'outburst' of carbonate platforms should be much late (in the Paleoproterozoic) based on the global distribution of sediment units.

Response of COMMENTS FROM REVIEWERS

Response to Reviewer #1

Reviewer #1 (Remarks to the Author):

Thanks for the opportunity to review this very interesting manuscript, the authors present good field observations and nice data on the Archaean North China Craton and the Great Oxidation Event with the global vision. The manuscript should be of general interest to a wide audience. It is well written and the figures are all extremely well-done and convey the message. I recommend publication after minor revision, with a few suggestions below, my concerns mainly include: (1) this manuscript showed a nice field observation in the Zanhuang mélange of the North China craton, but I would suggest the authors should improve the field interpretations a little bit, such as Figure 4, the authors might miss some important information among the blocks, and also some different foliations/boundaries; (2) I didn't see the carbonate in the field photos, maybe, the authors forget to identify them, however, it's such a really nice parametamorphic rock association in the NeoArchaean Zanhuang mélange; (3) In the Figure 1D, they should be foliations? but they were identified as like the bedding. Scales are need in the geological maps and all filed photos. I don't comment on the tectonic model discussion, it looks reasonable to me.

REPLY: Thank you for the great comments. We have considered your comments sincerely and addressed them in the revision to the text and figures as follows: (1) The interpretation of figure 4 is supported by our field observations and documentation, yet we revised it a bit to avoid misunderstanding of contact relationships. Changes to the figure are annotated in the new manuscript. (2) The carbonate field photos are shown in Figures 2d, 5e, 5g, and 7a, 7b, and 7c. (3) Considering they are high-grade metamorphic rock, the terminology 'foliation' which applies to metamorphic rock rather than 'bedding' to sedimentary rock is used in the figure. Scales are all now properly put in the maps and photos in the new manuscripts. Thank you for the insightful comments which helped us to improve the article a lot.

Response to Reviewer #2

NCOMMS-22-29586

Emergence of passive margins in accreting Archaean archipelagos signals continental stability and creation of pre-GOE oxygen oases

Peng and Kusky et al.

Review by Ross N. Mitchell (ross.mitchell@mail.iggcas.ac.cn)

This is an excellent idea with field data to support it about Archaean geology with implications for the leadup to the GOE. The accretion of ocean arc archipelagos is likely how many cratons formed—explaining nicely the granite–greenstone architecture of most Archaean cratons. The authors present the clever idea that the passive margins of Archaean archipelagos that accreted to form protocontinents would have, upon continental stabilization, subsided and created shallow water ecospace in which photosynthetic bacteria could have thrived and therefore eventually caused the tipping point of the GOE. Although their locality, North China craton, is just one, the authors are correct to point out that it is the *cumulative* effect of the cratons stabilizing globally that would have led to this increased ecospace over time, with later-forming cratons like North China being the proverbial “straw that broke the camel’s back”. The work is detailed and robust and the idea is compelling and fitting for publication in a general journal like Nature Communications. Although my comments below may seem critical, they are made only in an effort to improve an already strong and provocative paper. I would recommend MINOR REVISIONS.

REPLY: Thank you for the positive comments!

Major comments

(1) The pre-GOE oases, or pre-GOE “whiffs” of oxygen should not be taken as fact and the authors must admit somewhere that there is uncertainty about their existence. In fact, they are quite contentious and potentially dubious post depositional diagenetic artifacts (Slotznick et al., 2022). There is also depositional modeling inspired by the Archaean presence of redox-sensitive detrital grains of pyrite and uraninite (easily weatherable if free oxygen is available anywhere in the atmosphere) that casts theoretical doubt on the existence of pre-GOE “whiffs” of oxygen, suggesting the rise of oxygen must be younger than 2415 Ma (Johnson et al., 2014). I would therefore recommend the authors instead focus on the “leadup” or the “prelude” to the GOE. 2.5 Ga is not the start of the GOE. Some think the GOE was late as 2.33 Ga (Luo et al., 2016). Even if as old as <2415 Ma (Gumsley et al., 2017), the authors’ archipelago subsiding backsides are still not a “coincidence” with the GOE, but a prelude or leadup to it. Perhaps the delay can be explained in terms of subsidence creating sufficient accommodation space that shallow platform shelves covered enough areal extent to finally promote the full-fledged GOE. Furthermore, the more arcs that protocontinents accreted, the more continents would stabilize and such larger and higher continents would shed more sediment, making it more likely to form broad continental shelves to provide the ideal shallow water ecospace for photosynthetic bacteria. Also, it could be mentioned that sourcing rock-derived phosphorous as nutrients (which would be rich in apatite in basalts of ocean arcs) is a prerequisite for the GOE (Cox et al., 2018). Whereas evidence of photosynthesis may substantially pre-date the GOE (Nutman et al., 2016), nutrients were likely the rate-limiting reactant, which the authors’ model can nicely explain.

REPLY: We appreciate the critical and insightful comments of reviewer Ross Mitchell. You made good points which we largely agree with, and have modified the text as suggested throughout the text, as shown by the highlighted sections. Indeed, the notion of oxygen whiffs is contentious depending on the methods used to detect oxygen level, and the common agreement made by previous research, and our article as well, is merely that the oxygen oasis would remain in limited areas (as oases) if they do exist. The thing is, GOE would only be recorded when the concentration of oxygen level in the global atmosphere reaches to a threshold that allows oxygen to be widely involved in chemical reactions, which will make the timing of defined GOE delayed to the first emergence of oxygen oasis. The 2.4 to 2.3 Ga initiation time of the GOE^{2, 3, 4} does not conflict with our presumption, and is in agreement to the numerical model of oxygen level we referred to in the article⁵. The work here is meant to explain the transition between pre 2.5 Ga anoxic Earth and the 2.3 Ga oxidized planet. The point we try to make through the article is to provide a tectonic or geodynamic model linked to these oxygen oases and to imply the connection between continents formed by arc accretion and the global atmospheric changes by regional oxygen whiff accumulation. The general idea is that, by amalgamating arcs to form continents at around 2.5 Ga, the subsidence of protocontinent margins would contribute to oxygen formation, and as more continents became stabilized, the oxygen level would thus rise to the level and would trigger the GOE.

(2) In the profile description, it seems the island arc (graywacke) and backside sediments (calc-silicate) are conformably contacted? If so, why do the carbonates necessarily occur in a backside setting? These sediments can simply be part of the arc (arcs may have hiatus). Some Phanerozoic intra-oceanic island arcs have volcanoclastic rocks interbedded with the carbonate. If the island arc is in fault contact with the backside sediments, are these backside sediments comparable with the continental passive margin sediments? If they are, it means the island arc is in fault contact with both sides of the sediments. Then the simplest way to understand the arc-continent collision process is that the arc might have been thrust onto the continent (like the Macquarie Arc in SE Australia).\

REPLY: Thanks for the insightful comments. We would like to make it clear that the graywacke and underlying sandstone are interpreted to be products of immature continental margin costal sedimentation. And yes, they are in conformable relationship to the overlying calc-silicious rock. It is true that an intra-arc environment is capable of producing carbonate deposition, as in Taiwan, and the Banda Arc, etc. The differences are that in an intra-arc setting, the carbonate is interbedded within graywacke, which indicates back and forth regression and transgression settings or turbulent conditions. In our case, the graywacke does not reappear upon the carbonate sequence and there is a significant transition to a quiet and stable platform setting with details described in the text. It might be argued that the later deformation truncated the sequence, preventing the preservation of the later graywacke. But the sedimentation age constrained by the MDA in the carbonate-mudstone unit (ca. 2510 Ma) is so close to the collisional event (~2510-2500 Ma) that would leave fewer possibilities for further sedimentation. This is our evidence to support that the carbonate-mudstone represents the end of this sequence. In summary, our sequence indicates a transgressive and stable platformal environment, distinct from the regressive-transgressive and unstable intra-arc setting.

After that, at 2510-2500, the outboard arc was thrust upon the continent forming to the east, so our model is indeed consistent with your suggestions.

(3) Arc-continent collision process might be the direct contribution to any variations of the environment in this case (?). You have constantly new passive margins replacing the old ones. There is nearly no net increase of passive margin only from this process (assuming the length isn't changing as it's not discussed here). The continent is growing, but the passive margins are just constantly being replaced. The direct observations from the evidence shown in this paper are 1) arc-continent collision processes resulted in 2) continental growth/addition of carbonate. Thus, it would be clearer to state that arc-continent collision process is the key (?). The proposal of Phanerozoic arc-continent collisions in the tropics set Earth's climate state might be worth to be referred (Macdonald, Francis A., et al. Science 364.6436 (2019)).

REPLY: Thanks for pointing out this statement. Yes, you are right. The key is the continuous arc collision and amalgamation. The passive margin in arc-accretion setting is the result of arc amalgamation forming protocontinents, and the link between arc-accretion (tectonic) and the possible atmospheric change (sedimentation). Our study is the direct field evidence to indicate the influence of tectonic style to the planet's atmospheric changes.

Minor comments

By my count the current title is two words over the 15-word NC limit. I might suggest: Passive margins in accreting Archaean archipelagos signals continental stability and creation of pre-GOE oxygen oases

(I would avoid the word "emergence" for meaning "appearance of" or "rise of" passive margins given that the paper also discusses continental emergence, i.e., above sea level.)

Archaean should use UK spelling for this journal

REPLY: Thank you for pointing it out. We have revised the spelling throughout, and fixed the title to fit the word limit, and limit our use of the word emergence to refer to only emergence of continents above sea level to avoid confusion.

Abstract is quite good, but could be better. Some words to delete are suggested that may allow for a better re-write

REPLY: Done. Thanks.

Line 19 see comment about not using "emergence" unless continental emergence

REPLY: Fixed.

Line 19 late Archaean should not be hyphenated

REPLY: Fixed.

Line 22 passive-margin type

REPLY: Fixed.

Line 26 awkward wording "open-ocean back-side"?

REPLY: Fixed.

Lines 27 and 28 shallow water

REPLY: Fixed.

Line 28 photosynthetic microbes

REPLY: Fixed.

Line 29 still don't know what archipelago "back-sides" are? Also the backside (of a hill, for example) is an unhyphenated word.

REPLY: The front of the arc is the fore-arc, facing the trench, the backside of the arc, is marked by an open ocean or back arc basin. We have made sure this is clear in the text, and accompanying figure 9.

Line 29 represents

REPLY: Fixed.

Line 29 delete "environmental"; it is either redundant or even wrong (niches are ecological phenomena)

REPLY: Fixed.

Line 31 Aren't "initial" and "proto"(continental) kind of redundant?

REPLY: Right. Fixed.

Line 32 Archaean–Prot boundary most certainly isn't "early Earth", which is typically applied to Hadean and the earliest Archaean eras only. Delete and save some words for better abstract.

REPLY: Corrected.

Line 34-37 Quite a word salad for a first sentence! Do you really need to lead with the contentious onset of PT? Largely irrelevant by late NeoArchaean—when likely everyone in the world, with few or one exception, already expects PT would have been operational. Why not just jump into the contentious GOE and the even more contentious pre-GOE "whiffs" of oxygen

REPLY: This part of text is rewritten to get to the point.

Line 38 submerged

REPLY: Corrected.

Lines 34-40 short first paragraph (especially if you delete the essentially irrelevant onset of PT debate; previous comment); consider moving Lines 41-45 up to paragraph one.

REPLY: Fixed.

Line 45 “In this study, we report...” (NC formatting guidelines)

REPLY: Fixed, thanks

Line 45 passive-margin type shallow water sequence (these hyphenation corrections will not be repeated but should be corrected throughout)

REPLY: Fixed.

Line 46 northern China (North China not a formal term)

REPLY: Fixed.

Line 48 fuzzy is too colloquial for scientific writing; synonym uncertain/incomplete/etc.

REPLY: Fixed.

Line 49 Possessive with inanimate nouns is too colloquial for scientific writing; rewrite

REPLY: Fixed.

Line 52 nearly 100 or even 200 Myr in between the A–P boundary and the GOE....

REPLY: We have discussed this more clearly in the revision

Line 53 Results section should start here. Starting with a sub-section “Geologic setting.”

REPLY: Fixed.

Line 56 lowercase citations of figure panels in NC (fix throughout)

REPLY: Fixed.

Line 64 Southwest

REPLY: Fixed.

Line 66 use (refs. X) to avoid confusion with units (fix throughout when an ambiguity)

REPLY: Thank you for the comment. The comment is ambiguous, not sure what the reviewer is referring to. We, however, have tried to avoid confusion in the new manuscript.

Line 77 for numeric ranges, en dash (–) should be used and without spaces before/after (fix throughout)

REPLY: Fixed.

Line 278 upsection... and maybe listed the older age first to make more intuitive the upsection younging trend

REPLY: Fixed.

Line 280 transport

REPLY: Fixed.

Line 281 on this continental margin (since deposition is last in sequence)

REPLY: Fixed.

Line 287 define abbreviation

REPLY: Fixed.

Line 293 than the normal duration of an episode of sedimentation

REPLY: Fixed.

Line 292 such that the MDC

REPLY: Fixed.

Line 303 in the Supp...

REPLY: Fixed.

Line 310 The MDA

REPLY: Fixed.

Line 321 verify/validate/test (validify not a word)

REPLY: Fixed.

Line 325 unclear meaning of "above".. overlying sequence or aforementioned sequence?

REPLY: Fixed.

Line 334 close representation of the true depositional age (also no need to introduce an abbreviation used only once.. and on the *previous* page!)

REPLY: Fixed.

Line 354 avoid zombie compound nouns...the volcanics related to subduction initiation

REPLY: Fixed.

Line 357 Wilson cycle

REPLY: Fixed.

Line 362 sometimes fully hyphenated, sometimes not; be consistent

REPLY: Fixed.

Line 366 thermal-subsidence-related

REPLY: Fixed.

Line 368, 401 50 Myr later (duration not age)

REPLY: Fixed.

Line 436 Again, see Major Comment about timing of authors' mechanism and GOE

REPLY: Fixed.

Figs can be better.

REPLY: Fixed

Fig 1. keep the font consistent.

REPLY: Fixed.

Fig. 1A. the degree sign is missing (e.g., 100°E); the red square (for B) can be more obvious; keep the cross-section lines format consistent (a-b, c-d both with/without bars at the ends)

REPLY: Fixed.

Other figs: keep the upper/lower cases consistent for the units. For example, both CM and m were used; Line thickness of the legends; Keep the north arrows consistent; Use a big arrow for a map (like in Figs 1 and 3); For a cross-section, an arrow with a tail is more often used like Fig 7 better than Fig 4

REPLY: Thank you for pointing out all the incorrect and ambiguous expression in the text and figures. It is very helpful and appreciated! We have corrected the text and figures according to the above comments.

Cox, G. M., Lyons, T. W., Mitchell, R. N., Hasterok, D., and Gard, M., 2018, Linking the rise of atmospheric oxygen to growth in the continental phosphorus inventory: *Earth and Planetary Science Letters*, v. 489, p. 28-36.

Gumsley, A. P., Chamberlain, K. R., Bleeker, W., Soderlund, U., de Kock, M. O., Larsson, E. R., and Bekker, A., 2017, Timing and tempo of the Great Oxidation Event: *Proceedings of the National Academy of Sciences*, v. 114, p. 1811-1816.

Johnson, J. E., Gerpheide, A., Lamb, M. P., and Fischer, W. W., 2014, O₂ constraints from Paleoproterozoic detrital pyrite and uraninite: *Geological Society of America Bulletin*, v. 126, p. 813-830.

Luo, G., Ono, S., Beukes, N. J., Wang, D. T., Xie, S., and Summons, R. E., 2016, Rapid oxygenation of Earth's atmosphere 2.33 billion years ago: *Science Advances*, v. 2, p. e1600134.

Nutman, A. P., Bennett, V. C., Friend, C. R. L., van Kranendonk, M. J., and Chivas, A. R., 2016, Rapid emergence of life shown by discovery of 3,700-million-year-old microbial structures: *Nature*, v. 537, p. 535-538.

Slotznick, S. P., Johnson, J. E., Rasmussen, B., Raub, T. D., Webb, S. M., Zi, J.-W., Kirschvink, J. L., and Fischer, W. W., 2022, Reexamination of 2.5-Ga “whiff” of oxygen interval points to anoxic ocean before GOE: *Science Advances*, v. 8, no. 1, p. eabj7190.

Response to Reviewer #3

Reviewer #3 (Remarks to the Author):

This paper reports a suspicious NeoArchaean passive margin-type sequence from the North China craton. Tectonostratigraphic and detrital zircon analyses are approached to decipher its nature. It is concluded that this unit was sitting on the open-oceanic back-side of an amalgamating arc terrane in a shallow-water environment due to local thermal subsidence. If this is approved, it represents a stable environment that is distinct from the typical passive margins, but might be a specific environment signaling initial protocontinental maturity. This environment might provide a new opportunity for oxygenic lift prior to the Great Oxidation Event. This study is innovative - not only declaring a unique sequence at the initiation of continental maturity in the NeoArchaean, but also explaining the possible mechanism of subsidence of early continent to create shallow-water environment to be capable of early oxygenic life.

Thank you for the summary. We apply field mapping, including base maps from the

Geological Survey, tectonostratigraphic analysis, structural analysis, detrital zircon analysis, and analysis of the U-Pb and Pb-Pb ages of cross-cutting units, and the ages of late NeoArchaean metamorphism to make our interpretations of this Archaean sequence.

My major concern is the age of the key tectonostratigraphy in the paper. Although it has not been clarified directly in the paper (it should have had), it is quite clear that the key sequence includes the Gantaohu Group based on the geological map (Figure 1) and the description of its four units. There are many previous local and international publications for this group (e.g., Liu et al., 2012; Du et al., 2016). It is well-known that it is the Paleoproterozoic in age since its definition in 1960s (for example, there is an age of ~2.1 Ga for volcanics: Du et al., 2016). All these previous data on the group should be at least reorganized and evaluated.

REPLY: Thank you for your critical comment to help us clear up the possible confusion from future readers. We should have made it clarified in the previous submission, that we are dealing with the Archaean sequence, not the Proterozoic cover (Gantaohu Group).

According to the geological map and previous research work cited in both this work and the comments, the reported rock has well-constrained Archaean age and is distinguishable to the Proterozoic Gantaohu Group.

- 1) The Gantaohu Group is located about > 28 km northwest of our sequence, and rests unconformably over the Archaean Western Zhanhuang Domain (WZD, see Fig.1b) in our article. It is in no way same to the strongly deformed sequence reported in the manuscript that is normally considered as the 'Archaean basement' in research related to Gantaohu Group.
- 2) We are working on the Archaean basement, and our robust geochronology studied on both the sedimentary units, and the cross-cutting igneous units, indicate unequivocal Archaean age. These rocks were originally mapped as the Neoproterozoic Fangjiapu Formation in the Zhanhuang Group (or Tuanpokou Formation in Fuping Group), see ref. 23, 24 in the main text.
- 3) Although the rock assemblage is broadly similar, the sequence of Gantaohu Group is related to a Proterozoic rifting event according to previous research, 400 million years younger than the arc-continent collisional event correlated to this sequence (line 245-254).

Although the relationships are certain, we would like to apologize to the reviewer for not making that clear in the initial submission. In order to clarify the misunderstanding, we have revised our Figure 1b, to show the location of the Gantaohu Group in relationship to the rocks we described, and have added a new couple of sentences, stating the official formation names of these rocks, and refer to the 1:50,000 scale geological maps. As stated in the text, we used these geological survey maps of the Archaean basement, as base maps for our own maps and defined a stratosection, based on their previous work, as stated on lines 83-89, and 92-99. The statement is:

These rocks were originally mapped as the Fangjiapu Formation in the Zanhuang Group (or Tuanpokou Formation in Fuping Group)^{23,24}, but in following sections, we use a lithostructural nomenclature based on our new results. Parts of the Western Zanhuang Domain are unconformably overlain by flat-lying sedimentary and volcanic rocks of the 2.1 Ga Gantaohe Group^{22,23} (Fig. 1b) showing that major deformation in the Zanhuang massif was over by that time.

Another major concern is the interpretation on the tectonic environment of the sequence. It is quite depending on the author's own model on the tectonic evolution of the region, as well as their interpretation on the age of the Gantaohe Group (see above). Based on the sedimentary facies of the group, as well as the chemical and geochronological data already reported, it has been proposed that the Gantaohe Group was developed on the continental crust, either in an intra-continental rifting environment (e.g., Du et al., 2016 and many other researches) or a back-continental arc environment (e.g., Liu et al., 2012).

REPLY: The Gantaohe Group relates to a much later event later than the 2.5 Ga arc-continent collision^{1, 6}. We agree with the mechanism proposed by previous research on the genesis of Gantaohe Group. But this paper is not about the Gantaohe Group, it is about the underlying late Archaean basement.

In addition, both the mechanism for the subsidence of the so-called 'NeoArchaean' passive marginal basin and its link with the Great Oxidation Event are not adequately supported by the data provided in the paper (hardly proved but just stated). Is there any sign of thermal subsidence of the region? Is there any fossil or tracer to prove the flourish of oxygenic life in the sequence?

REPLY: Our age data on the NeoArchaean age of this sequence are robust, supported by our detrital zircons, Archaean metamorphic rims on zircons, and crystallization age of cross-cutting late Archaean igneous rocks.

Like many other Archaean metasedimentary rock, the sequence is highly deformed and metamorphosed, so fossils and tracers are difficult to be preserved in this condition. However, by considering the genesis of this thick carbonate sequence, we can decipher the presence of oxygen generating microorganisms. The transgressive nature of the sequence indicates a subsidence of the protocontinental margin.

Our work here, as well as that of the Geological Survey of Hebei Province, for more than two decades, indeed provides a solid framework for our interpretation, and is meant to provide a tectonic or geodynamic model linked to these oxygen oases and to imply the connection between continents formed by arc accretion and the global atmospheric changes by regional oxygen whiff accumulation at the same time frame. The general idea is that, by amalgamating arcs to form continents at around 2.5 Ga, the contemporary subsidence of the protocontinent margins would form environmental habitats suitable for organisms to thrive, and contribute to oxygen formation, and as more continents get to stabilized, the oxygen level would thus rise to the level and would trigger the GOE.

With the above concerns in mind, I think this innovative work is rather inadequate at this stage. I would be happy to see more evidence to depict it – if it has been proved, it would have provided a totally new concept to the research field, which with no doubt would be revolutionary.

REPLY: Thank you for the comments. The question you raised are addressed in detail below, but we need to clarify again, that we are not describing the Gantaohu Group in this work, we are describing and interpreting the Archaean basement.

Du, L., Yang, C., Wyman, D. A., Nutman, A. P., Lu, Z., Song, H., Zhao, L., Geng, Y., and Ren, L., 2016, Age and depositional setting of the Paleoproterozoic Gantaohu Group in Zhanhuang Complex: Constraints from zircon U–Pb ages and Hf isotopes of sandstones and dacite: *Precambrian Research*, v. 286, p. 59-100.

Liu, C., Zhao, G., Liu, F., Sun, M., Zhang, J., and Yin, C., 2012, Zircons U-Pb and Lu-Hf isotopic and whole-rock geochemical constraints on the Gantaohu Group in the Zhanhuang Complex: Implications for the tectonic evolution of the Trans-North China Orogen: *Lithos*, v. 146-147, p. 80-92.

We do not cite these papers, since they are about the Proterozoic Gantaohu Group, and our work is about the Archaean basement, not the Gantaohu Group. These papers do NOT include our described sequence as part of the Gantaohu Group. In fact, the first paper argues that since the Gantaohu Group is virtually undeformed, it was deposited after any collisions, which is quite consistent with our work.

Abstract:

Line 19-21: this statement on the significance of ‘the emergence of stable shallow-water continental platforms’ is rather arbitrary.

REPLY: Thank you for pointing it out. We have corrected it to be more appropriate now.

Line 26: how do you know it was on ‘the open-ocean back-side’? If it was on the open-ocean back-side, we won’t call it a ‘passive margin’?

REPLY: Thank you for the question. The tectonic setting is based on our regional tectonic synthesis. Our demonstration on the passive margin can be referred to lines 91-160, and 343-349. We note that our example, and some of the others are not “traditional Wilson Cycle” passive margins, where a previously existing large continent rifts, thermally subsides, then has a passive margin developed on it. This is an accretionary orogen, where the transgressive shallow water sequence developed on the back arc side of very recently amalgamated arcs. Related revised text is on lines 354-369, and 371-378.

Line 28: it is likely that there was no indication of ‘microbes’ in the stratigraphy (e.g., there was even no stromatolite in the carbonates).

REPLY: Good question. Considering the highly deformed and metamorphized nature of this rock suite, finely laminated stromatolites are unlikely to be preserved. The deformation of the carbonate-mudstone unit is indicated in Fig. 7.

Line 48: 'this data' should read 'these data' or 'the dataset'.

REPLY: Corrected.

Line 53-59: many geologists suggested that the eastern craton and western craton were collided at 1.9-1.8 Ga, rather than 2.9-2.5 Ga. Please at least mention this option.

REPLY: Thank you for raising this question.

The 1.9-1.8 Ga collision has been proven incorrect by many papers in various journals, including in this journal⁷. The proposal of a 1.9 or 1.8 collision with the suture in this zone is clearly inconsistent with basic geological facts, for instance, the 2.1 Ga Gantaohe Group is virtually undeformed, and sitting on the site of the proposed 1.8 Ga suture in those studies, proving that there was no 1.8 Ga collision in this area (for instance, see the paper by Du et al., above, that you ask us to cite).

We, and others, discuss this debate in many papers. However, it is not the subject of this paper. Here, we are describing the 2.56-2.50 Ga shallow water sedimentary sequence, and its involvement in a 2.50 Ga arc continent collision, which is well established. We, nevertheless, have addressed the 1.8 metamorphism overprinting in our work very clearly in Supplementary Discussion 2, lines 138-144. The 1.8 Ga metamorphic event is recorded nearly everywhere in the craton, and not related to a specific event in this location.

Further, we are not saying the Eastern and Western Blocks collided at 2.5 Ga as a single event. The eastern Block likely formed by arc amalgamation by 2560 Ma, then the next arc of the Central Belt collided at 2510-2500 Ma. Then many other things were accreted until 1.8 Ga, as you mention, and we cite relevant papers, especially references 7-22, 25, 27, 28, 30, 35, and 67. The 1.8Ga topic is not the focus of this paper, and it has been addressed adequately in these other papers.

Line 60-63 = '... causing deformation and metamorphism between 2.7-2.55 Ga to form protocontinental nuclei...' – however, there are hardly metamorphism ages of 2.7-2.55 Ga in the North China craton. Actually, the widespread metamorphism in this craton was happened most likely after 2.55 Ga. BTW, 'nucleii' should read 'nuclei'?

REPLY: That's not true, actually there are many reports on metamorphism around 2.7 – 2.55 Ga^{8, 9, 10}, which are overprinted by the strong 2.5 Ga collision events, then a second overprinting event around 1.85 Ga. Spelling mistake is corrected, thanks.

Line 72-80: the regional geology should start with the units defined by local workers, and then you may specify your tectonic interpretation for each unit.

REPLY: Good points. In the new manuscript, we have cited regional geology of local workers in this manuscript, referred to line 83-89 and ref. 23, 24. Yet, we have made a clear litho-structural description, not an assumed layer cake stratigraphy for highly deformed and metamorphosed rocks. The lithological description can be found on lines 92-156, and the contact relationships are on lines 163-170 and 184-193.

Line 83-90: I see no evidence of allochthonous bodies/units in the Zanhuang region. Based on what you define them as 'allochthonous' units? Ages different from the very small local basement? The tectonic contact among units? No strong speculation has been stated here or hereafter. The whole speculation is suspicious.

REPLY: The allochthonous bodies/units in the Zanhuang region is reported in many previous works^{6,7}, including in this journal. We think the reviewer is confused, thinking that we are reporting work from the unconformably overly 400-million-year younger undeformed basin sequence of the Gantaohu Group, which is, as stated, not allochthonous.

The allochthon is a forearc mélange complex, completely unrelated to the continental passive margin setting of our autochthonous sequence⁷. The tectonic contact between the allochthon the autochthon is a series of thrust faults^{7, 11}, with the closest fault to our sequence marked out in Fig. 7. The tectonic contact between the units within the sedimentary sequence is a major shear zone, very similar to the main nappe-bounding shear zones of the Alps¹¹ and Appalachians. The allochthonous nature of the overlying nappes is well-confirmed by numerous studies, based on solid field observation and mapping work, with good constraints from previous work on the arc-continent collision event and from our work on the pre-collisional sedimentation. Please see the publications cited above, including Zhong et al., 2021, and 2022, from this journal, and *Geology*.

Line 102-104: 'The quartz diorite ... characteristic of mid-crust intrusions'. I do not aware that quartz diorite intrusives should be mid-crustal. Please show the observations and evidence.

REPLY: Thank you for the question. The characteristic of mid-crust intrusions is that, under a higher temperature in the mid-crust, an unclear boundary and fuzzy contact is expected between the intrusion of small volume of magma body to the country rock, which matches to our field observation.

The evidence is shown in our detailed map, and photos, and data, in Figure 3, and in the related text. Considering that it is not essential to this work, related statement is deleted due to ambiguity.

Line 106-108: which episodes of deformation? Late Archaean or Paleoproterozoic (episodes as revealed by both zircon U-Pb and amphibole Ar-Ar ages)?

REPLY: The deformation is considered the arc-continental collision at around 2.50-2.45 Ga^{6, 7, 11, 12}. It is very clearly demonstrated in the text. Again, this question is based on the wrong assumption that we are dealing with the Gantaohu Group.

Line 109: 'cover'? Which is/are the 'cover'? Please specify.

REPLY: The term "cover" refers to the sedimentary sequence covering the magmatic complex that is considered as continental basement. It is a standard geologic term, but we have corrected the expression to make it clearer, thanks.

Line 111-144: what are the thickness of each unit (without counting the deformation)?

REPLY: Good question. The original thickness is hard to measure as we are not certain on the shortening effect brought by the deformation during the arc-continent collision. The original thickness is, thus, not discussed in the article (not the main target of this manuscript) but may require further work to determine. The current thicknesses are discussed, and documented in many places (e.g., Fig. 7).

Line 121-124: all these sequences were metamorphosed into high-amphibolite to granulite facies without any preservations of grading. How do you know they are graywackes rather than shales to siltstones?

REPLY: We determine the protolith by the mineral composition. Shale is likely metamorphized into mica schist and siltstone into quartzite, while graywacke would be observed as mica quartzofeldspathic gneiss. We do describe rock fragments and graywacke protoliths, graded beds, and sedimentary structures. In general, siltstones and shales do not have large rock fragments of immature detritus. These are shown in Figs. 2, 5, 6

Line 131: 'shallow-water mudstone'? Based on what?

REPLY: We determined it as shallow water based on shallow water nature of the whole sequence. For example, sudden uplift or rapid subsidence would bring dramatic changes in lithology, inconsistent to the gradual transition observed in the field. The reasoning of the statement is referred to line 146-150.

Line 134-135: 'low energy environment'?

REPLY: The higher part of unit 3 are thick, pure sediments of arenite, siltstone, carbonate, and mudstone, which is considered deposited in low energy, quiet environment. The reasoning of the statement can be found on lines 146-150.

Line 136-138: The arenite comprised by quartz, feldspar and biotite has 'minor' compositional differences with quartzite??

REPLY: Quartzite in our sequence is composed of fine-grained equigranular quartz, while

arenite is composed of medium- to coarse-grained quartz, feldspar, and biotite, indicating different protoliths and depositional environments.

Line 143: how do you know the 'marbles' are originally 'limestone'?

REPLY: Although deformed, the marble-metapelite unit has interbedded shales, which indicates the protolith to be sedimentary rather than igneous, as in carbonatites. At the microscopic scale, we document the major composition of the marble is recrystallized calcite, whose chemical composition is CaCO_3 . The finely interbedded rocks are shales, as illustrated in Figs. 2, 5,6, and 7. Thus, we identify this sedimentary rock with a major composition of CaCO_3 as limestone.

Line 146-151: why this metabasalt-featured unit allochthonous slice rather than a part of the stratigraphy? The whole sequence was well-known as the Gantaohu Group in the literature. Although it has undergone stronger deformation in the middle of the Zhanhuang region, it shows quite consistent sequences with those west to the region with less deformation. It is well-known to be a Paleoproterozoic intracontinental sequence.

REPLY: This is NOT the Gantaohu Group! The location of Gantaohu Group is nowhere near where our sequence is. Also, Gantaohu Group is a typical intracontinental basin sequence it has no relation to the deformed belt under the Archaean thrust belt that we describe here.

The metabasalt unit is in thrust fault contact with the sedimentary rock, as shown in Fig. 7. Below, to make it clearer, we paste some figures showing our documentation of the thrust faults, and nappes, of the Archaean sequence⁷, which is part of the Archaean melange belt, cut by Archaean dikes and pluton.

Fig. 2 Profile of Black Rock Temple and Buddha nappes (both named after local names); lower-hemisphere equal area stereographic projections of D_1 and D_2 axial surfaces, fold hinge lines, and lineations. Structural data for Black Rock Temple nappe is above dashed line, for the Buddha nappe, below line. Mapping was performed at a scale of 1:666.

The fold-thrust nappes thrust over the passive margin contrast in rock type, age, structure, metamorphism, with the Gantaohu Group, as pictured in this following figure⁵.

[redacted]

We hope we have made it clear enough that the sequences are NOT related. We are NOT describing the Gantaohu Group.

Line 149: It should be noticed that the Archaean basement in the COB has varied ages from 2700-2500 Ma.

REPLY: We demonstrated the age of the Basement in line 179-183, and discuss the varied ages, clearly in the text.

Line 152-155: The statement here is arbitrary.

REPLY: We made it clearer and follow the standard tectonic terminology now, thanks.

Line 157-159: Observation is needed to show an overthrust allochthonous unit. A tectonic/fault contact does not guarantee an allochthonous origin.

REPLY: You are right, one fault contact does not guarantee such a relationship. However, our statement is constrained by detailed field mapping in this region. Along with the fault shown in this work, there is a nappe structure, with a 2.7 Ga forearc mélange complex, including 2.7 Ga picritic-boninitic volcanics, thrust upon the 2.5 Ga continental margin sequence⁷. Our claim is based on 20 years of detailed field work and structural analysis, reported in the literature, in which we define the autochthon and the allochthon^{6, 7, 11}. The observations are documented in more than 40 peer-reviewed papers, some of which we cite here.

Lines 165-166: Sure thing. The basement is older than 2.5 Ga; while the strata was middle Paleoproterozoic in age.

REPLY: The reviewer thinks we are dealing with the Proterozoic Gantaohé Group. We are not. They are far apart, and totally different in structural style, rock assemblages, tectonic genesis, as described above in many places.

The sequence is constrained to be 2560-2510 Ma, i.e., late Archaean, consistent with previous Geological Survey and other mapping, as cited. The middle Paleoproterozoic sequence is the later Gantaohé Group inside Western Zhanhuang Domain, >28 km away from our sequence¹. Our sequence underwent the 2.5 Ga arc-continent collision, 2.5 Ga metamorphism, and intrusion by 2.5 Ga undeformed dikes and plutons, explained in the structural analysis section in the text, which indicates the sequence was deposited prior to 2.5 Ga. This is NOT the Gantaohé Group deposited 400 million years later. The age of the basement is demonstrated in line 179-183.

Line 169-176: yes, there was probably an episode of metamorphism at the late Archaean, but the major episode was the late Paleoproterozoic.

REPLY: There are metamorphic events in both late Archaean and the late Paleoproterozoic^{9, 10, 13, 14}. In our studies, we find two metamorphic events, one at 2.5Ga recording an arc-continent collision (impossible if this were the 2.1 Ga sequence) and another at 1.8Ga. 1.8Ga was the major metamorphic event in most of the North China Craton, as it was related to continent/continent collision in the northern margin of the North China Craton, and when the craton joined the Columbia Supercontinent.

Line 182-186: Again, it is reasonable that the deformation of the basement and the Gantaohé Group (unit 1-4) is different.

REPLY: We have stated the differences between Gantaohé Group and our sequence. This is NOT the Gantaohé Group. We have added this illustration in 84-87 to make it clear.

Line 190-247: sorry I did not read it in quite detail. It is really hard to follow as the major deformation here, or at least the deformation as revealed by Ar-Ar ages is the late

Paleoproterozoic. Without distinguishing the Archaean and Paleoproterozoic episodes of deformation, it is really hard to evaluate the description.

REPLY: Good point. However, Ar-Ar dating can be easily reset by thermal events because of its low closure temperature. The age of the deformation is determined by the crosscutting relationship, where ~2.50 Ga undeformed granitoid cut across the intensely deformed mélange, as reported in articles published in the past 10 years throughout the eastern margin of the COB. The overprinting metamorphism of the marble-metapelite unit is reported by¹⁰. Detailed constraints on tectonic events can be found in ^{6, 7, 11, 12}.

Line 266-269: the error of these ages are quite large, over 20 Ma or 30 Ma. There are no obvious differences between these age groups to discriminate their provenances.

REPLY: Right, the error can be 20 or 30 Ma. So, in our study we analysis >53 data for each sample, with careful observation and filtering to make sure the data are representing the magmatic formation age of each zircon. We interpret the provenance based on these sufficient data, among the high-quality data, using a variety of statistical methods that are well-established. The demonstration on the provenance discrimination methods can be referred to line 275-278. The differences of the ages in different samples can be referred to line 285-291. In the Supplementary Discussion 1, line 105-108, we have stated clearly that the data from detrital zircon is carefully filtered and properly interpreted to age calculation, and it is, therefore, valid in tectonic discrimination. This is described in great detail in the Supplementary Information, and represents the best age constraints, ever, on the autochthonous Archaean platform of the Zhanhuang mélange belt. It is not the Gantaohé Group.

Line 270-286: I admit that there could be a successive thermal/igneous event in the region during the Late Archaean; then how can you distinguish ages between domains with lead loss and domains not, especially considering their large errors of singular ages.

REPLY: Yes, you are right. Singular data with lead loss can be easily misinterpreted. That is the reason why we apply sufficient data to demonstrate the distribution pattern of the ages instead of using single zircons for interpretation. The diagram with distribution pattern is shown in Fig. 8, and the calculation methods to eliminate errors of single data are explained in detailed in the Supplementary Discussion 1, line 43-53. That is also why we also apply geological relationships of cross cutting Archaean dikes to show that the sequence is Archaean. The crosscutting relationship is described in main text, line 263-266 and shown in Fig. 2b, and its age constraints can be referred to lines 265-268, 330-346, and in the Supplementary data.

This confusion may be largely based on the reviewer believes we are dealing with the Proterozoic Gantaohé Group, which is much younger than the rock that we reported in this study.

Line 287-335: all these analyses and interpretation is based on the assumption that the

sequence was the late Archaean in (depositional) age. But that was not the case, the volcanics (metabasalts) and other rocks show clear evidence that the rocks are middle Paleoproterozoic in age (e.g., Du et al., 2016; Liu et al., 2012).

REPLY: Again, this confusion is based on the reviewer believes we are dealing with the Proterozoic Gantaohu Group. We are not.

Line 321-325: it analyzed the age of a crosscutting metamorphosed mafic dyke, which gives a U-Pb zircon age of 2507 Ma. I highly doubts whether this is an age of crystallization or an age from inherited zircon grains, and hence, it is inadequately proved that this ~2507 Ma age is a key to constrain the deposition age of the tectonostratigraphy.

REPLY: We understand your doubts. It is apparently possible that mafic intrusion would have inherited zircon. That is why we have CL images provided in Supplementary Fig. 2e. In the figures we have shown the distinct differences between the inherited zircons (which have much older ages) and magmatic zircons. Please see the description in the supplementary data.

Line 338-340: But the whole basement was not exhumed until the Late Paleoproterozoic; how can it be the exposed coast in the Archaean.

REPLY: The age issue has been addressed very clearly with multiple constrains, which can be referred to 'Geochronology constraints' section The statement that the Archaean basement exhumed in the late Paleoproterozoic is irrelevant to our work on the Archaean basement. The assumption of not being exhumed until the Paleoproterozoic is unsupported. Again, this confusion is based on the reviewer believes we are dealing with the Proterozoic Gantaohu Group

Lines 358-361: 'Thus, the results strongly suggest the autochthonous passive margin sequence was deposited concurrently with accretion of multiple arcs, with intervening oceans of different ages, in an accretionary orogen, similar in style to the Altai of central Asia'. All the ages provided in the paper are within-error quite similar, and nothing of the kind can be supported.

REPLY: We disagree. The arc accretionary setting is defined by regional tectonic studies and detrital zircon age distribution pattern, which is well explained in the text. We have ensured our data and interpretation meet with the quantity and quality, precision, and accuracy, which makes our statement valid.

Line 374-376: what is the evidence for 'thermal subsidence' of the region in the Late Archaean?

REPLY: In the results part, we demonstrated a transgressive sequence, meaning the land is sinking relative to sea level. In the discussion part, we exclude the possibility of external effect of the subsidence. It is thus only thermal subsidence would be the reason of the

mechanism. The deduction can be referred to line 379-397.

Line 436-437: 'The sudden out-burst of biogenic carbonate platforms globally around 2.5 Ga ...'. Where does this statement come from? I would not show the dataset here, but the 'outburst' of carbonate platforms should be much late (in the Paleoproterozoic) based on the global distribution of sediment units.

REPLY: An outburst is never an absolute terminology. The outburst in this scenario is relative to poor bioactivities in early Archaean. Our presumption is based of global data in Archaean biogenic carbonate platform, which is shown in Fig.10. The figure clearly supports our claim.

Thank you for your review, and we are sorry you thought we were working on the Gantaohu Group, which we are not, and apparently led to most of the statements of doubt or confusion above.

References

1. Peng P, Yang S, Su X, Wang X, Zhang J, Wang C. Petrogenesis of the 2090 Ma Zhanhuang ring and sill complexes in North China: a bimodal magmatism related to intra-continental process. *Precambrian Research* 303, 153-170 (2017).
2. Johnson JE, Gerpheide A, Lamb MP, Fischer WW. O₂ constraints from Paleoproterozoic detrital pyrite and uraninite. *GSA Bulletin* 126, 813-830 (2014).
3. Luo G, Ono S, Beukes NJ, Wang DT, Xie S, Summons RE. Rapid oxygenation of Earth's atmosphere 2.33 billion years ago. *Science Advances* 2, e1600134 (2016).
4. Gumsley AP, et al. Timing and tempo of the Great Oxidation Event. *Proceedings of the National Academy of Sciences* 114, 1811-1816 (2017).
5. Lyons TW, Reinhard CT, Planavsky NJ. The rise of oxygen in Earth's early ocean and atmosphere. *Nature* 506, 307-315 (2014).
6. Wang J, et al. Structural relationships along a NeoArchaean arc-continent collision zone, North China craton. *GSA Bulletin* 129, 59-75 (2017).
7. Zhong Y, et al. Alpine-style nappes thrust over ancient North China continental margin demonstrate large Archaean horizontal plate motions. *Nature communications* 12, 1-15 (2021).
8. Kusky T, et al. Insights into the tectonic evolution of the North China Craton through

comparative tectonic analysis: A record of outward growth of Precambrian continents. *Earth-Science Reviews* 162, 387-432 (2016).

9. Wu C, Zhou Z, Zuza AV, Wang G, Liu C, Jiang T. A 1.9 – Ga mélange along the northern margin of the North China craton: Implications for the assembly of Columbia supercontinent. *Tectonics* 37, 3610-3646 (2018).
10. Xiao D, et al. NeoArchaean to Paleoproterozoic tectonothermal evolution of the North China Craton: constraints from geological mapping and Th-U-Pb geochronology of zircon, titanite and monazite in Zanhuang Massif. *Precambrian Research* 359, 106-214 (2021).
11. Zhong Y, Kusky TM, Wang L. Giant sheath-folded nappe stack demonstrates extreme subhorizontal shear strain in an Archaean orogen. *Geology* 50, 577-582 (2022).
12. Wang J, et al. Petrogenesis and geochemistry of circa 2.5 Ga granitoids in the Zanhuang Massif: Implications for magmatic source and NeoArchaean metamorphism of the North China Craton. *Lithos* 268, 149-162 (2017).
13. Xiao L-L, Liu F-L, Chen Y. Metamorphic P–T–t paths of the Zanhuang metamorphic complex: Implications for the Paleoproterozoic evolution of the Trans-North China Orogen. *Precambrian Research* 255, 216-235 (2014).
14. Xiao L-L, Chen M-H. Metamorphic Age Comparison and Its Implications between the Zuoquan and Zanhuang Complexes in the Central North China Craton, Based on LA-ICP-MS Zircon U–Pb Dating. *Minerals* 9, 780 (2019).

Reviewer #1 (Remarks to the Author):

Thanks for the revision, all looks nice and it will be a great contribution on the global Archaean geology. It should be accepted in this present.

Reviewer #2 (Remarks to the Author):

380547_1

Passive margins in accreting Archaean archipelagos signals continental stability promoting pre-GOE oxygen oases
Peng et al.

Second review by Ross N. Mitchell (ross.mitchell@mail.iggcas.ac.cn)

The authors have done a sufficient job in making revisions and further justifications of reviewer comments with which they disagreed. The revised Introduction is much stronger. I continue to be in favor of publication of this important piece of work pending MINOR REVISIONS.

The authors should cite and incorporate a machine learning approach to Earth's oxygen evolution recently published in Nature Communications¹. Most notably, the identification of two "Archaean oxidation events" (AOEs) at ca. 3.0 and 2.5 Ga would seem to support the authors' model that passive margins forming for the first time on protocontinents were important in the leadup to the GOE. As Chen et al. ¹ write, "the GOE was the culmination of an extended period of transient oxidation." Another recent paper ("Carbonates before skeletons"²) supports the authors' model, showing a large number of Late Archaean carbonates compared to the very few Early Archaean occurrences.

1 Passive margins "of" accreting Archean archipelagos...

19 Boring/dull to start a NC paper with "Previous work..."; Rewrite, something to the effect of: "Significant changes in tectonic style and climate occurred from late Archaean to early Proterozoic" (also saves some extra words for later in the Abstract, if needed)

21 by "oxygenic life", I believe the authors mean "photosynthetic life", or could say "oxygenic photosynthesis"

23 spell out "million years ago" for broad NC audience (with words saved above)

85 "the Tuanpokou Formation in the Fuping Group"

162 "Structural relationships."

192 "in-situ meter-scale"

256 "Geochronology constraints"

287 "implies" and delete comma

313 "Supplementary Text"

343 something missing in this sentence; needs to be written near the end

344 only use c. for approximate ages/dates; so "~50 Myr lifespan"

445 "indicates that such a process played a pivotal role in promoting the GOE"

449 the authors may consider citing a new paper that indicates a post-Archaean transition from bottom-up to top-down mantle convection, or simply “changing styles of mantle convection”³

498, 511, 520, 541, 549 suggest moving such details (transect X-X; location shown in Fig. xx; Supp Table xx, etc.) out of the bold figure headings and can be specified in the following figure descriptions

564 “Connection between protocontinent emergence in the Archaean and biogenic carbonate distribution” more succinct figure title

530 space missing

1 Chen, G. et al. Reconstructing Earth’s atmospheric oxygenation history using machine learning. *Nature Communications* 13, 5862, doi:10.1038/s41467-022-33388-5 (2022).

2 Cantine, M. D., Knoll, A. H. & Bergmann, K. D. Carbonates before skeletons: A database approach. *Earth-Science Reviews* 201, 103065, doi:https://doi.org/10.1016/j.earscirev.2019.103065 (2020).

3 Mitchell, R. N., Brown, M., Gernon, T. M. & Spencer, C. J. Evolving mantle convection from bottom up to top down. *The Innovation* 3, doi:10.1016/j.xinn.2022.100309 (2022).

Reviewer #3 (Remarks to the Author):

I would apologize for not being able to evaluate this revision in time. To avoid further delay, I would like to just focus the major concern of this manuscript – the age of the key units being stated in the paper. I would thank the authors to point out that these units were once mapped as the Fangjiapu Formation (this term was first used in 1961 but was abandoned later). With no doubt, there are also a couple of other popular editions, for example, the local 1:20 000 map released in 1996 called it the Honghe Formation (of the Zhanhuang) low - grade sequence. Nevertheless, no matter what it has been called, the age of this sequence is crucial for the paper – that is exactly what I have concerned in the original review. Here is a couple of examples with ages being reported: The detrital zircon ages suggest a maximum age of ca. 2200 Ma (Li et al., 2016; Zhang et al., 2019). Yet there is no good age for the volcanics in the sequence (ca. 2.3 Ga: e.g., Li et al. (2016)). I agree that more data are absolutely needed to give a better constrain (the whole units were metamorphosed), but please present evidence or better declaration if the authors think a part of this sequence is the Archean in age.

Figure 1. Simplified geological map of the region (Tang et al., 2016).

Figure 2. The cross-section of the sequence in the central Zhanhuang (Zhang et al., 2019).

Figure 3. Some representative ages of the sequence (Zhang et al., 2019).

See Figures 1-3 in review attachments.

References:

Li, S.-S., Santosh, M., Teng, X.-M., and He, X.-F., 2016, Paleoproterozoic arc-continent collision in the North China Craton: Evidence from the Zhanhuang Complex: *Precambrian Research*, v. 286, p. 281-305.

Tang, L., Santosh, M., Tsunogae, T., and Maruoka, T., 2016, Paleoproterozoic meta-carbonates from the central segment of the Trans-North China Orogen: Zircon U–Pb geochronology, geochemistry, and carbon and oxygen isotopes: *Precambrian Research*, v. 284, p. 14-29.

Zhang, F., Wang, Y.-B., Du, L.-L., Yang, C.-H., Ayers, J. C., and Yuan, H.-Q., 2019, The Neoarchean-Paleoproterozoic volcanic-sedimentary rocks in the Zhanhuang Complex, North China Craton: Petrogenesis and implications for tectonic evolution: *Precambrian Research*, v. 328, p. 64-80.

Reviewer #3 Attachments; Figure 1, Figure 2, Figure 3 on the following three pages.

[redacted]

Second Revision

Response of COMMENTS FROM REVIEWERS

Reviewer #1 (Remarks to the Author):

Thanks for the revision, all looks nice and it will be a great contribution on the global Archaean geology. It should be accepted in this present.

REPLY: Thank you for the positive comments!!

Reviewer #2 (Remarks to the Author):

380547_1

Passive margins in accreting Archaean archipelagos signals continental stability promoting pre-GOE oxygen oases

Peng et al.

The authors have done a sufficient job in making revisions and further justifications of reviewer comments with which they disagreed. The revised Introduction is much stronger. I continue to be in favor of publication of this important piece of work pending MINOR REVISIONS.

REPLY: Thank you for the constructive comments during the review!

The authors should cite and incorporate a machine learning approach to Earth's oxygen evolution recently published in Nature Communications¹. Most notably, the identification of two "Archaean oxidation events" (AOEs) at ca. 3.0 and 2.5 Ga would seem to support the authors' model that passive margins forming for the first time on protocontinents were important in the leadup to the GOE. As Chen et al. ¹ write, "the GOE was the culmination of an extended period of transient oxidation." Another recent paper ("Carbonates before skeletons"²) supports the authors' model, showing a large number of Late Archaean carbonates compared to the very few Early Archaean occurrences.

REPLY: Thank you for the constructive comment. It is a really good point that you brought up, and we are incorporating them into our revised manuscript, improving our Figure 10.

1 Passive margins "of" accreting Archean archipelagos...

REPLY: Thank you for the comment. We intend to demonstrate the locality by using 'in' rather than 'of'.

19 Boring/dull to start a NC paper with "Previous work..."; Rewrite, something to the effect of: "Significant changes in tectonic style and climate occurred from late Archaean to early Proterozoic" (also saves some extra words for later in the Abstract, if needed)

REPLY: The part has been rewritten. Really appreciate the effort.

21 by “oxygenic life”, I believe the authors mean “photosynthetic life”, or could say “oxygenic photosynthesis”

REPLY: Yes, thanks for pointing it out. We have fixed it.

23 spell out “million years ago” for broad NC audience (with words saved above)

REPLY: Fixed.

85 “the Tuanpokou Formation in the Fuping Group”

REPLY: Fixed.

162 “Structural relationships.”

REPLY: Fixed.

192 “in-situ meter-scale”

REPLY: Fixed.

256 “Geochronology constraints”

REPLY: Fixed.

287 “implies” and delete comma

REPLY: Fixed.

313 “Supplementary Text”

REPLY: Fixed.

343 something missing in this sentence; needs to be written near the end

REPLY: Fixed.

344 only use c. for approximate ages/dates; so “~50 Myr lifespan”

REPLY: Fixed.

445 “indicates that such a process played a pivotal role in promoting the GOE”

REPLY: Fixed.

449 the authors may considering citing a new paper that indicates a post-Archaeon transition from bottom-up to top-down mantle convection, or simply “changing styles of mantle convection”³

REPLY: Thanks for the suggestion, we liked this paper, it came out after our last submission. We have updated and incorporated it into the discussion in line 448-449.

498, 511, 520, 541, 549 suggest moving such details (transect X-X; location shown in Fig. xx; Supp Table xx, etc.) out of the bold figure headings and can

be specified in the following figure descriptions

REPLY: Fixed.

564 “Connection between protocontinent emergence in the Archaean and biogenic carbonate distribution” more succinct figure title

REPLY: Fixed.

530 space missing

REPLY: Fixed.

1 Chen, G. et al. Reconstructing Earth’s atmospheric oxygenation history using machine learning. Nature Communications 13, 5862, doi:10.1038/s41467-022-33388-5 (2022).

2 Cantine, M. D., Knoll, A. H. & Bergmann, K. D. Carbonates before skeletons: A database approach. Earth-Science Reviews 201, 103065, doi:<https://doi.org/10.1016/j.earscirev.2019.103065> (2020).

3 Mitchell, R. N., Brown, M., Gernon, T. M. & Spencer, C. J. Evolving mantle convection from bottom up to top down. The Innovation 3, doi:10.1016/j.xinn.2022.100309 (2022).

REPLY: We thank Reviewer 2 for the time and effort on these constructive comments. We are incorporating them into the latest revision.

Reviewer 3

I would apologize for not being able to evaluate this revision in time. To avoid further delay, I would like to just focus the major concern of this manuscript – the age of the key units being stated in the paper. I would thank the authors to point out that these units were once mapped as the Fangjiapu Formation (this term was first used in 1961 but was abandoned later). With no doubt, there are also a couple of other popular editions, for example, the local 1:20 000 map released in 1996 called it the Honghe Formation (of the Zhanhuang) low - grade sequence. Nevertheless, no matter what it has been called, the age of this sequence is crucial for the paper – that is exactly what I have concerned in the original review. Here is a couple of examples with ages being reported: The detrital zircon ages suggest a maximum age of ca. 2200 Ma (Li et al., 2016; Zhang et al., 2019). Yet there is no good age for the volcanics in the sequence (ca. 2.3 Ga: e.g., Li et al. (2016)). I

agree that more data are absolutely needed to give a better constrain (the whole units were metamorphosed), but please present evidence or better declaration if the authors think a part of this sequence is the Archean in age.

Reply: We are sorry that reviewer 3 did not have time to go through all the detail of our data and the discussion on data analysis before giving the comments. We would like to restate two points: the naming of the stratigraphy units and the quality of the ages referred by the reviewer.

For naming issue, we abandoned the various old names assigned to these rocks, and their correlations with other units, as they were based on poor study on Archean orogenic events, resulting in the interpretation of the thrust sheets and nappes in an accretionary orogen (see for example the recent papers by^{1,2}) as a once continuous stratified sedimentary column stretching across tectonic units that were initially widely separated in space in time.

For the ages referred by reviewer 3, the poor quality and the inappropriate interpretation are the reason why we didn't refer to these works. The Zhanhuang complex records a complex history of collision and accretion, and in such case, caution is needed in data interpretation. All the ages should be interpreted with integrated field relationships and accurate data analysis. Only when rigorous and systematic research is conducted, can the interpretation be reliable.

Our age is constrained by 290 valid detrital zircon ages with at least 53 for each, all with detailed description of field lithology and structural relationship. While the ages reported in those two papers are poor in quality, lack of field description, and thus are geological meaningless. Please see the attached "Letters to the Editor" published by Precambrian Research^{3,4}, where we pointed out the sub-standard quality and quantity of data used both by SS Li et al (2016)⁵ and L Tang et al (2016)⁶, which the authors of this group (led by M Santosh) keep propagating through the literature. Their interpretation also contracts the basic field relationships that we have established from years of detailed mapping in the Zhanhuang massif^{1,2,7,8,9}.

After the Archean history, that we focus on in this paper under consideration in Nature Communications, the region indeed experienced new widespread sedimentation and magmatism at about 2.1 Ga, which is well-described and analyzed in the recent papers by Peng Peng¹⁰ and others.

Despite our clear disagreement about the age of these deformed strata, we thank Reviewer 3 for checking on this – one of the main driving forces for this research was to correct the idea that has wormed into the literature, from the papers by Tang et al. (2016)⁶ and Li et al. (2016)⁵ based on low-quality data, that these are simply Proterozoic, virtually undeformed strata. With our hundreds of new analyses, plus analyses of cross-cutting igneous rocks, we show that these rocks are clearly Archean, which has great implications for the start of the GOE, life, and how our planet has evolved, so as scientists, we insist on using high-quality data, before making models.

References:

Li, S.-S., Santosh, M., Teng, X.-M., and He, X.-F., 2016, Paleoproterozoic arc-continent collision in the North China Craton: Evidence from the Zhanhuang

Complex: *Precambrian Research*, v. 286, p. 281-305.

Tang, L., Santosh, M., Tsunogae, T., and Maruoka, T., 2016, Paleoproterozoic meta-carbonates from the central segment of the Trans-North China Orogen: Zircon U–Pb geochronology, geochemistry, and carbon and oxygen isotopes: *Precambrian Research*, v. 284, p. 14-29.

Zhang, F., Wang, Y.-B., Du, L.-L., Yang, C.-H., Ayers, J. C., and Yuan, H.-Q., 2019, The Neoproterozoic-Paleoproterozoic volcanic-sedimentary rocks in the Zanhuang Complex, North China Craton: Petrogenesis and implications for tectonic evolution: *Precambrian Research*, v. 328, p. 64-80.

Reviewer Attachment 1. Simplified geological map of the region from (Tang et al., 2016).

[redacted]

We hereby attach our map (greatly simplified on left) from Wang et al. (2017)⁷, based on our earlier work, in our discussion of Tang et al. (2016)^{4, 6}, compared with map reviewer 3 refers to.

The map reviewer 3 provides has many errors, we will not use it. First, please note that the longitudes put this map in the middle of the Pacific Ocean way off the coast of Japan. Note the careless incorrect assignment of lithological units as mica schist and gneiss, where it is K-feldspar granite, and granite, where they are layered mafic complexes and flat lying sediments. Please see the discussion by Wang et al. (2017)⁴, on the bad data and science of this paper.

Attachment 2. The cross-section of the sequence in the central Zhanhuang (Zhang et al., 2019).

[redacted]

The profile presented in Attachment 2 is rough and oversimplified, with numerous errors of rock type identification, structure, age, relationships between units, and is close to either meaningless or misleading. We spent some years to make the profiles present in this contribution, and in our companion paper (attached), and in our previous research paper¹. One example of this, is our Buddha nappe, please compare our part of the entire profile (attached below) with reviewer 3's suggestion, which we refute:

Comparison of the profile section we are asked to cite (above) and ours for the same section (below, Buddha nappe)¹:

[redacted]

[redacted]

Comparison of the section we are asked to cite (left) with ours, for the same section(right)².

The profile Reviewer 3 sends as attachment 2, has been re-done, and is in review at present.

Reviewer Attachment 3. Some representative ages of the sequence (Zhang et al., 2019).

[redacted]

To help clear up the age dispute, we hereby plot together the metasediments reviewer 3 refers to, another geochronological study on marble-metapelite unit, and our sequence into the figure 1 geological map. In this way, we compare all the data, and elaborate the reason why the age reviewer 3 suggested is incorrect.

[redacted]

The 2200 Ma age¹¹ reviewer 3 refers to comes from sample Z54-1 in this figure. The ages from this sample are highly discordant, which is unqualified to use for depositional age constrain. Moreover, no field observation or description of lithology of this specific sample are reported in the original article to support their notion, and the way they illustrate structural relationships (in reviewer's attachment 2, shown above) is rough and oversimplified. Subsequently, these data are in no way 'representative' of the regional sedimentary sequence.

From another sequence, the Z82-1 quartzite from Zhang et al. (2019)¹⁰, is from a suite of sedimentary rocks north of our sequence (Figure above), which yielded "a single age peak of ~0,3 DZ p_mahn_m Zgr rhng[^]k Z[^] ^[^] khn_i || { 'Cb[^] nk[^] 4), which has no contradiction to our result.

In summary, none of the data reviewer 3 refers to^{5,6,11} is properly interpreted in this complicated setting, and thus in no way suggests the 2.3 Ga age of our sequence.

Reference cited:

1. Zhong Y, *et al.* Alpine-style nappes thrust over ancient North China continental margin demonstrate large Archean horizontal plate motions. *Nature Communications* **12**, 1-15 (2021).
2. Zhong Y, Kusky TM, Wang L. Giant sheath-folded nappe stack demonstrates extreme subhorizontal shear strain in an Archean orogen. *Geology* **50**, 557-582 (2022).
3. Wang J.P., Deng H., Kusky T.M., and Polat A. Comments to "Paleoproterozoic arc-continent collision in the North China Craton: Evidence from the Zhanhuang Complex" by Li *et al.* (2016), *Precambrian Research* **286**, 281-305 (2017).
4. Wang, J.P., Deng, H., Kusky, T.M., and Polat, A., Comments to "Paleoproterozoic meta-carbonates from the central segment of the Trans-North China Orogen: Zircon U-Pb geochronology, geochemistry, and carbon and oxygen isotopes" by Tang *et al.*, 2016, *Precambrian Research* **284**, 14-29. *Precambrian Research* **294**, 344-349 (2017).
5. Li S.S. *et al.* Paleoproterozoic arc-continent collision in the North China Craton: Evidence from the Zhanhuang Complex. *Precambrian Research* **286**, 281-305 (2016).
6. Tang L. *et al.* Paleoproterozoic meta-carbonates from the central segment of the Trans-North China Orogen: Zircon U-Pb geochronology, geochemistry, and carbon and oxygen isotopes. *Precambrian Research* **284**, 14-29 (2016).
7. Wang J, *et al.* Structural relationships along a Neoproterozoic arc-continent collision zone, North China craton. *Geological Society of America Bulletin* **129**, 59-75 (2017).
8. Wang J, *et al.* Petrogenesis and geochemistry of circa 2.5 Ga granitoids in the Zhanhuang Massif: Implications for magmatic source and Neoproterozoic metamorphism of the North China Craton. *Lithos* **268**, 149-162 (2017).
9. Wang J, Kusky T, Polat A, Wang L, Deng H, Wang S. A late Archean tectonic mélangé in the Central orogenic belt, North China craton. *Tectonophysics* **608**, 929-946 (2013).
10. Peng P, Yang S, Su X, Wang X, Zhang J, Wang C. Petrogenesis of the

2090 Ma Zanhuang ring and sill complexes in North China: a bimodal magmatism related to intra-continental process. *Precambrian Research* **303**, 153-170 (2017).

11. Zhang F. *et al.* The Neoproterozoic-Paleoproterozoic volcanic-sedimentary rocks in the Zanhuang Complex, North China Craton: Petrogenesis and implications for tectonic evolution. *Precambrian Research* **328**, 64-80 (2019).
12. Xiao D. *et al.* Neoproterozoic to Paleoproterozoic tectonothermal evolution of the North China Craton: constraints from geological mapping and Th-U-Pb geochronology of zircon, titanite and monazite in Zanhuang Massif. *Precambrian Research* **359**, 106214 (2021).

Attachment 1: Detailed response to the age issue

To help clear up the age dispute, we hereby plot together the sample locations of metasediments to which reviewer 3 refers, along with locations of other geochronological studies on the marble-metapelite unit, and our sequence, on Figure 1, our geological map. In this way, we compare all the data, and elaborate on the reasons why the age reviewer 3 suggested is incorrect.

[redacted]

Figure 1. The sampling location in the article reviewer 3 refers to, another geochronological study on marble-metapelite unit (Xiao et al., 2021), and our sequence.

Sample ZH-9 is a dolomite-calcite marble reported by Tang et al. (2016) but has highly suspicious ages with huge analytical errors (figure 2), leading us (and Wang et al., 2017a) to conclude that the data and conclusion based upon that data are untenable. The authors only present 13 data points (excluding many more), which is not enough to rule out contamination and anomalous data, and very far from enough for a statistically valid detrital zircon study. In our study, we present 290 analyses, sufficient for a robust statistical analysis. Furthermore, the CL images of this sample (figure 7d in the original article) show significant hydrothermal alteration (Figure 3; outlined by a white solid line), which overlaps with their analytical spots (yellow circles), showing that their younger age result is clearly a result of contamination, and has no meaning as a depositional age constraint.

This was the subject of the “Letter to the Editor” by JP Wang et al (2017) stating that the data presented by L Tang et al was low quality and meaningless. The Letter is attached.

[redacted]

Figure 2. The age plot of ZH-9 from Tang et al. (2016), originally as figure 9 c-d. Only 13 analytical spots were used, which is insufficient for any meaningful detrital zircon geochronology study.

[redacted]

Figure 3. Interpretation of ZH-9 from (Tang et al., 2016) shows the contamination of young alteration to the analysis, yielding spurious ages that are mixtures between the original age and the alteration age. The white solid line outlines the metamorphic/metasomatized rim of the zircon. The blue dotted lines show the possible igneous zonation in the cores. Yellow circles are the analytical spots from the original figure.

The 2200 Ma age in Zhang et al. (2019) to which reviewer 3 refers (Figure 4), comes from sample Z54-1. The ages from this sample are highly discordant, which is unqualified to use as a constraint to estimate depositional age. Moreover, no field observations or descriptions of the lithology of this specific sample are reported in the original article to support their notion, and the way they illustrate structural relationships is rough and unclear (figure 5, left) compared to the more accurate ones of all the detailed structures and lithologies from our studies (Figure 5 right). In contrast, another study carried

out in a similar location (Xiao et al., 2021), with well-described field relationships and a reliable amount of analysis (Figure 6), yield similar age constraint to our study, that is Archean ages for sedimentation and magmatism, late Archean metamorphism, followed by a second metamorphic event at ~ 1.85 Ga.

[redacted]

Figure 4. The Z54-1 age diagram from Zhang et al. (2019), originally as figure 8 a-b.

[redacted]

Figure 6. Age plots of the marble unit from Xiao et al. (2021), originally as figure 8 e-f.

Zhang et al. (2019) report results from sampling a suite of sedimentary rocks north to our sequence (figure 1), from which sample Z82-1 yields “a single age peak of 2.5 Ga without any younger age groups” (Zhang et al., 2019; figure 4), which has no contradiction to our result.

In summary, none of the data reviewer 3 refers to (Tang et al., 2016; Li et al., 2016; Zhang et al., 2019) are properly interpreted in this complicated setting, and thus in no way suggests a 2.3 Ga age of our sequence. Conversely, our study is conducted following the procedures of rigorous scientific conduct and reporting. In this work, we have presented detailed reports on field lithology and relationships, well-constructed chronological sequences, including maximum depositional age constraints from 290 analyses, statistically sufficient (refer to lines 295-327, and 333-344), as well as crosscutting relationships of intrusions (refer to lines 329-333) and of overprinting deformation (refer to lines 245-254) at 2.50 Ga, using accurate figures and age constraints from previous research (Wang et al., 2017a, b; Zhong et al., 2021, 2022), and the new results we report.

Reference cited:

- Li, S.S., Santosh, M., Teng, X.M. and He, X.F., 2016. Paleoproterozoic arc-continent collision in the North China Craton: Evidence from the Zhanhuang Complex. *Precambrian Research*, 286, pp.281-305.
- Tang, L., Santosh, M., Tsunogae, T. and Maruoka, T., 2016. Paleoproterozoic meta-carbonates from the central segment of the Trans-North China Orogen: Zircon U–Pb geochronology, geochemistry, and carbon and oxygen isotopes. *Precambrian Research*, 284, pp.14-29.
- Wang, J., Kusky, T., Wang, L., Polat, A., Deng, H., Wang, C. and Wang, S., 2017a. Structural relationships along a Neoproterozoic arc-continent collision zone, North China craton. *Bulletin*, 129(1-2), pp.59-75.
- Wang, J., Kusky, T., Wang, L., Polat, A., Wang, S., Deng, H., Fu, J. and Fu, D., 2017b. Petrogenesis and geochemistry of circa 2.5 Ga granitoids in the Zhanhuang Massif: Implications for magmatic source and Neoproterozoic metamorphism of the North China Craton. *Lithos*, 268, pp.149-162.
- Xiao, D., Ning, W., Wang, J., Kusky, T., Wang, L., Deng, H., Zhong, Y. and Jiang, K., 2021. Neoproterozoic to Paleoproterozoic tectonothermal evolution of the North

China Craton: constraints from geological mapping and Th-U-Pb geochronology of zircon, titanite and monazite in Zanhuang Massif. *Precambrian Research*, 359, p.106214.

Zhang, F., Wang, Y.B., Du, L.L., Yang, C.H., Ayers, J.C. and Yuan, H.Q., 2019. The Neoproterozoic-Paleoproterozoic volcanic-sedimentary rocks in the Zanhuang Complex, North China Craton: Petrogenesis and implications for tectonic evolution. *Precambrian Research*, 328, pp.64-80.

Zhong, Y., Kusky, T., Wang, L., Polat, A., Liu, X., Peng, Y., Luan, Z., Wang, C., Wang, J. and Deng, H., 2021. Alpine-style nappes thrust over ancient North China continental margin demonstrate large Archean horizontal plate motions. *Nature communications*, 12(1), pp.1-15.

Zhong Y, Kusky TM, Wang L. Giant sheath-folded nappe stack demonstrates extreme subhorizontal shear strain in an Archean orogen. *Geology*. 2022 May 1;50(5):577-82.

Attachment 2: Previously published Letters to the Editor (of *Precambrian Research*), related to the data reviewer 3 refers to.

We hereby attach our published Letters to the Editor (of *Precambrian Research*), that previously pointed out the poor quality of data being used by M Santosh's two students, Li Tang, and S.S. Li, and several of our published papers on the ages of the rocks on this profile. If requested we could also send our paper, currently in review, of our detailed structural transect of the entire Zanhuang profile.

These two letters are:

1. Wang J.P., Deng H., Kusky T.M., and Polat A. Comments to "Paleoproterozoic arc-continent collision in the North China Craton: Evidence from the Zanhuang Complex" by Li et al. (2016), *Precambrian Research* **286**, 281-305 (2017).
2. Wang, J.P., Deng, H., Kusky, T.M., and Polat, A., Comments to "Paleoproterozoic meta-carbonates from the central segment of the Trans-North China Orogen: Zircon U-Pb geochronology, geochemistry, and carbon and oxygen isotopes" by Tang et al., 2016, *Precambrian Research* **284**, 14-29. *Precambrian Research* **294**, 344-349 (2017).